# Quantitative determination of the spatial distribution of components in single cells with CellDetail

Tanja Schuster [1] ✉, Amanda Amoah [1,2], Angelika Vollmer[1], Gina Marka[1], Julian Niemann[1], Mehmet Saçma [1], Vadim Sakk[1], Karin Soller[1], Mona Vogel[1], Ani Grigoryan [1], Meinhard Wlaschek[3], Karin Scharffetter-Kochanek [3], Medhanie Mulaw [4] & Hartmut Geiger [1] ✉

The distribution of biomolecules within cells changes upon aging and diseases. To quantitatively determine the spatial distribution of components inside cells, we built the user-friendly open-source 3D-cell-image analysis platform **Cell Det**ection and **A**nalysis of **I**ntensity **L**ounge (CellDetail). The algorithm within CellDetail is based on the concept of the dipole moment. CellDetail provides quantitative values for the distribution of the polarity proteins Cdc42 and Tubulin in young and aged hematopoietic stem cells (HSCs). Septin proteins form networks within cells that are critical for cell compartmentalization. We uncover a reduced level of organization of the Septin network within aged HSCs and within senescent human fibroblasts. Changes in the Septin network structure might therefore be a common feature of aging. The level of organization of the network of Septin proteins in aged HSCs can be restored to a youthful level by pharmacological attenuation of the activity of the small RhoGTPase Cdc42.

A distinct spatial distribution of biomolecules within a cell is tightly linked to the proper function of a cell. Polarity is a special form of a spatial distribution of biomolecules. Polarity describes an asymmetric clustering of components within the cell space (Fig. 1a). For the commonly used visual assessment of polarity, a virtual line separating the cell into two halves is chosen in a way to maximize for polarity. A distribution with several, completely dispersed biomolecule clusters or with two biomolecule poles with approximately the same amount opposite to each other is regarded as apolar (Fig. 1a). Polar distributions of proteins are for example found in yeast, epithelial cells, neurons and in young but less so in old hematopoietic stem cells (HSCs, Fig. 1b, polar distribution of the polarity protein Cdc42)[1–6]. Not only proteins but also mRNAs and whole cell organelles can show a polar distribution[2,7]. A dysregulated polar distribution of biomolecules has been reported in diseases like microvillus inclusion disease, Crohn's disease, chronic lung diseases, Huntington disease, microcephaly, epilepsy and cancer[1,2,8–16].

Usually, the position of biomolecules in cells is identified by (immuno)fluorescence. There are several approaches available for analyzing the spatial distribution of a fluorescent signal within a cell, like visual assessment[5,17–22], neural networks, angle measurements to morphological features[23–25], intensity switches in regions of interest[20,26,27], barycenter calculations[28] or a generalized approach describing cell and nucleus morphology by coefficients and defining common states[29]. In general, each of these approaches shows shortcomings like subjectivity (visual assessment, neural networks), categorization instead of continuous values (visual assessment, neural networks), reduced information content (intensity barycenter calculation, ratio of intensity, generalized approach) or require clearly identifiable morphological features in cells (angle measurements, generalized approach).

[1]Institute of Molecular Medicine, Ulm University, Ulm, Germany. [2]Terry Fox Laboratory, BC Cancer Research Centre, Vancouver, BC, Canada. [3]Department of Dermatology and Allergic Diseases, Ulm University, Ulm, Germany. [4]Unit for Single-Cell Genomics, Ulm University, Ulm, Germany. ✉e-mail: tanja.schuster@uni-ulm.de; hartmut.geiger@uni-ulm.de

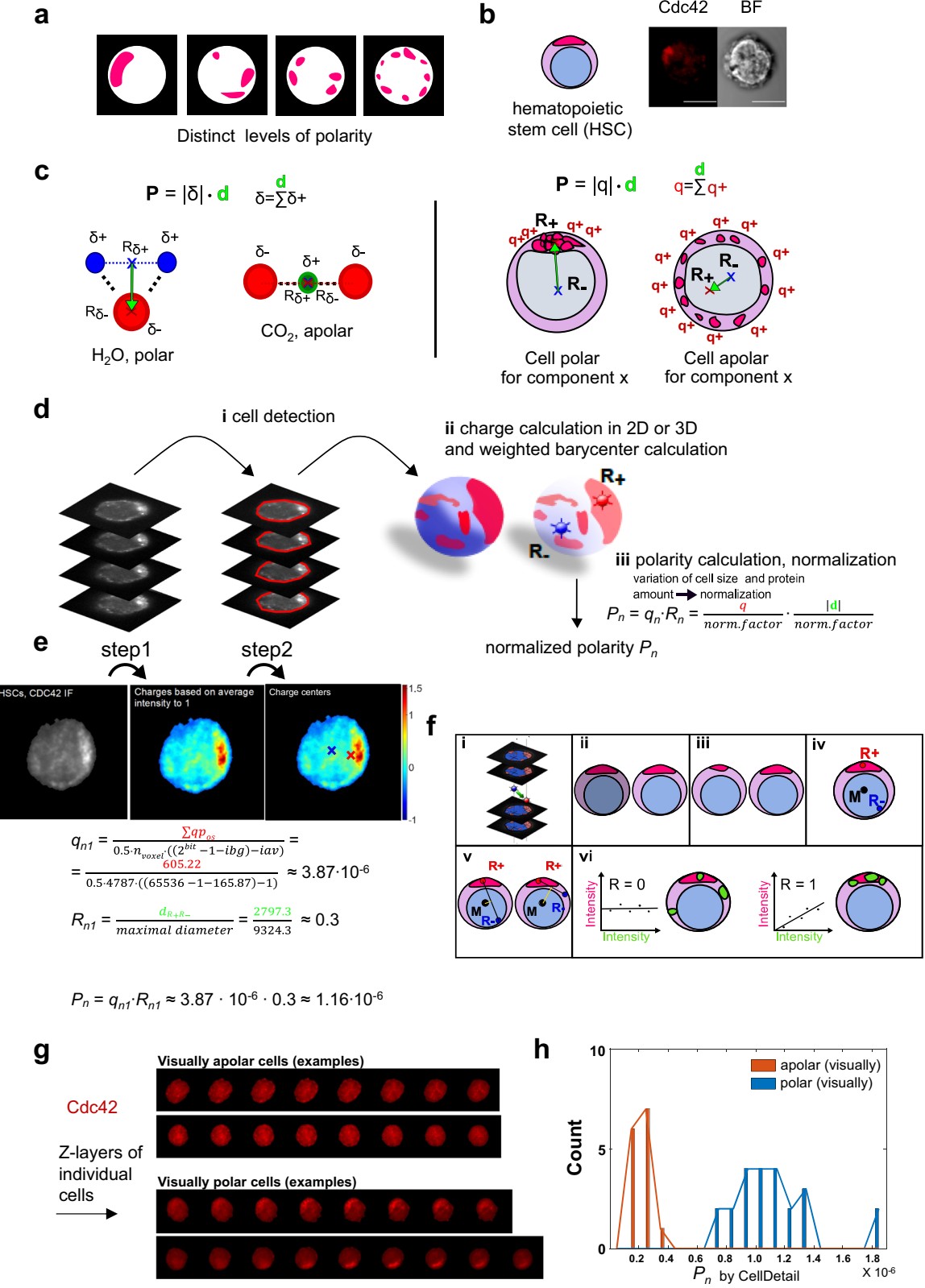

Automated quantitative analyses of the spatial distribution of components/proteins within isotropic cells like HSCs that do not show clearly identifiable morphological features therefore remain a challenge. This is why visual assessment of the spatial distribution of the fluorescence is still the most widely used approach. To overcome such shortcomings, we built the platform **Cell Det**ection and **A**nalysis of **I**ntensity **L**ounge (CellDetail), which allows for the quantitative evaluation of the distribution of biomolecules in cells, including isotropic cells like HSCs, at a 3D resolution. CellDetail comes with a graphical user interface (GUI). It is easy to use as only minor manual adjustments need to be chosen to adapt the program for a specific use. CellDetail works fast, as it does not make extensive use of segmentation tools, and no neural networks need to be trained. CellDetail was validated by (i) artificially generated data and (ii) confirmation of

**Fig. 1 | Spatial distribution, algorithm and CellDetail. a** The level of polarity of the distribution of a component in a cell can adopt multiple distinct states and can thus be regarded as a continuous variable. **b** HSC with a polar distribution of the protein Cdc42 (red) (left: schematic, right: confocal IF image and bright field (BF), scale bar = 5 μm). **c** Left top: Equation to calculate the dipole moment (polarity) of chemical molecules. Left low: $H_2O$ is a highly polar molecule, while $CO_2$ is a highly apolar molecule. ($\delta$ = charges, **R** = charge centers, **d** = distance, green arrow, **P** = dipole moment). Right top: Equation to calculate the dipole moment of a component within a cell ($q$ = charges, **R** = charge centers, **d** = distance, green arrow, **P** = dipole moment). Charge centers of the red component in the cell are shown as red (positive charge center) and blue (negative charge center) crosses. **d** Sequential steps for quantifying polarity via the value $P_n$ as output parameter in 3D IF images. **e** Example calculation of $P_n$ for the protein Cdc42 in HSC (here single 2D layer).

Step1 shows the conversion of intensity values into charge values by subtracting the average cell intensity, step2 shows the charge-weighted barycenters. More detail on these steps can be found in Supplementary F Algorithm. (ii) $P_n$ is calculated based on the (i) normalized charge ($q_{nl}$) and the normalized distance ($R_{nl}$) between the charge centers. **f** Output parameters of CellDetail: (i) polarity quantification, (ii) amount of the biomolecule, (iii) spatial volume of biomolecule, (iv) vectors between middle of cell and positive/negative charge-weighted centers, (v) grade of constriction of biomolecule charge centers, (vi) Pearson correlation analyses ($R_{Pearson}$) for inter-biomolecule behavior analyses. **g** Examples of z-stacks (3D) of HSCs visually apolar and polar for the distribution of the protein Cdc42 in HSCs (out of 14 apolar and 23 polar used in (**h**)). **h** Histogram of $P_n$ values for the protein Cdc42 for 14 HSCs visually apolar for Cdc42 and 23 HSCs visually polar for Cdc42. Source data are provided as a Source Data file.

previously reported altered spatial distributions of the polarity proteins Cdc42 (a small Rho-GTPase) and Tubulin in aged HSCs and upon their rejuvenation[2].

The quantitative analysis of the level of polarity by CellDetail provides more in-depth information on the spatial distribution of a biomolecule compared to a binary (yes/no) output. Polarity itself is indeed a quantitative value as indicated by intermediate states between visually identified polar and apolar states (Fig. 1a). The "analog" output of CellDetail allows for correlative analyses on co-distributions of biomolecules and for the analysis of the nature of multi-protein complex networks of proteins. Septins are cytoskeletal proteins known for polarity regulation and maintenance and also cell stiffness[30]. The spatial organization of Septin7 is controlled by the level of activity of Cdc42[31]. The activity of Cdc42 is elevated in old HSCs and in other types of aged cells[2,32]. It is thus likely that the organization of the Septin network, and thus an important player in the organization of the cytoskeletal order, is altered in old HSCs[33]. By employing now CellDetail and a novel complex IF panel for staining multiple Septins in single cells, we demonstrate an overall less polar organization of the Septins and their network in old HSCs which was reverted to a young like, polarized distribution upon pharmacological attenuation of Cdc42 activity in aged HSCs. Similar to old HSCs, human senescent (aged) fibroblasts showed also less polar distribution of Septins, implying that changes in the distribution of Septins and change in their network might be a common feature among aged cells.

## Results

### The dipole moment is the underlying concept of the algorithm of CellDetail

The polarity quantification algorithm of CellDetail is modeled on the molecular dipole moment **P** usually applied to molecules like $H_2O$ (polar) or $CO_2$ (apolar) (Fig. 1c, left). The dipole moment is defined as the absolute value of charge $\delta$ (total positive, or respectively, negative charge) multiplied with the distance vector **d** between positive and negative charge center ($R_{\delta+}$ blue cross, $R_{\delta-}$ red cross): $\mathbf{P} = |\delta| \cdot \mathbf{d}$. This concept was transferred to fluorescence images of biomolecules in cells (Fig. 1c, right). The distance vector between charge-weighted positive and negative charge centers, multiplied by the absolute value of charge, provides the value for the dipole moment which is considered as a measure for the polarity of the distribution of the component under investigation. Regions of low biomolecule density form the negative charge center $R_-$ (blue cross), regions with a high biomolecule density form the positive charge center $R_+$ (red cross). By multiplying the total amount of positive charges $q$ with the vector **d** (= distance vector, green) between the charge centers, the resulting dipole moment **P** is a value for the polarity of the distribution of the fluorescence intensity (aka the biomolecule) within the cell. For 3D analyses, the charge centers as well as the positive charges are calculated across all layers together, not per single layer. See also Supplementary Information F, Algorithm.

CellDetail follows a sequential order of analysis steps (Fig. 1d): (i) cell detection is performed on either an image stack or single images. Single layer or z-stacked tiff images of single cells with a fluorescence signal are used as input. Cell detection is included in CellDetail. It is based on Otsu's method for automatized thresholding. Then, voxels are separated into background voxels and cell voxels (cell detection). Subsequently, the average intensity of background voxels is calculated and subtracted from the cell voxel intensities to obtain background corrected voxels for each cell and the intensity of the voxels is normalized to the average cell intensity per voxel being 1. Next, the average intensity value of voxels is calculated and subtracted from voxel intensities leading to positive and negative voxel values, resulting in the charges $q_+$ and $q_-$, then the charge centers $R_+$ and $R_-$ and subsequently the dipole moment **P** is calculated in 3D and normalized (Fig. 1e and Supplementary Information F, Algorithm).

In chemical molecules like $H_2O$ (polar) or $CO_2$ (apolar), both the distance between charge centers as well as the amount of charge are not variable within and among molecules (aka each molecule looks alike). In contrast, each cell is distinct in size and shape, so maximal distance of charge centers is also dependent on the cell size. The amount of the charge is further dependent on the overall amount of the fluorescence signal (total protein of interest content) and the microscope settings (bit-depth with which the image was acquired). Normalizations of the distance (Supplementary Information F Fig. S9) and charge and protein content (Supplementary Information F Fig. 10) are therefore required to allow to compare polarity values among cells. To this end, the absolute value of the dipole moment **P** is normalized (Supplementary Information F) to result in the normalized dipole moment $P_n$. Supplementary Information F Table S2 and Figs. S11–S15) provide detailed information on the normalization process. Decision trees on normalization options to guide choices of normalization option for distinct types of images/distributions are provided. CellDetail includes two options for distance normalization, five options for charge normalization and two options for protein content normalization. $P_n$ is the central output parameter of CellDetail (Fig. 1e). CellDetail works in general in 3D, but can also be used for single layer images (2D).

CellDetail provides the following output parameters (Fig. 1f): (i) the normalized dipole moment of a a biomolecule, $P_n$, (ii) the amount/intensity of the biomolecule in the cell, (iii) the spatial volume of the biomolecule in the cell, (iv) the vectors between charge centers $R_+$, $R_-$ and the position **M** of the center point of the cell which are required for the determination of correlations of biomolecule 1 to biomolecule 2 position like angles and distances, (v) a distance describing the constriction of positive and negative charge center to a subcell region or across a cell to see whether polarity is constricted to a subcell structure and (vi) co-localization parameters like voxel intensity-based Pearson correlation coefficients across channels/molecules. CellDetail has a graphical user interface (GUI) (https://github.com/xyq91/CellDetail-TS for downloading CellDetail, Supplementary Information H Manual). Output files of CellDetail are.txt, .xls and .mat files. A full list and

description of all output parameters is listed in the CellDetail manual (Supplementary Information H Manual). Robustness of CellDetail concerning brightness variation, SNR variation, oversaturation and shape differences was confirmed (Supplementary Information C) and the distribution function of $P_n$ was mathematically derived (Supplementary Information I).

CellDetail correctly separated distinct HSCs based on 3D polarity/apolarity that were visually identified to be polar and apolar for Cdc42 and used as the reference data set (Fig. 1g, h). More importantly, in-depth benchmarking analyses that involved other currently available polarity analysis tools or algorithms demonstrated that CellDetail performed in general with a higher level of accuracy in separating image layers of HSCs visually scored for being polar or apolar for Cdc42, while being similar to the accuracy of the barycenter method (Supplementary Information B, Fig. S2 and Table S1).

### Polarity of Cdc42 and Tubulin in young, old and old rejuvenated HSCs

HSCs from young mice show a high frequency of cells polar for the polarity protein Cdc42 or for Tubulin[2,34]. Polarity in HSCs is tightly linked to an asymmetric division of HSCs. An asymmetric division ensures proper stem cell homeostasis[19]. Upon aging, the frequency of HSCs with a polar distribution of these proteins decreases, and aged HSCs are primarily apolar. Apolarity is a critical hallmark of aged stem cells[2,35–37].

The identification of polarity in HSCs has been so far primarily performed by visual examination of 2D or 3D images (Fig. 2a, Supplementary Movie 1 and Supplementary Movie 2)[2]. We set out to quantify the change in the polarity values $P_n$ of Cdc42 and Tubulin in 3D (stained via established IF protocols[2]) in individual young and old HSCs. This data might also serve as additional validation test for CellDetail. We analyzed z-stacked confocal microscopy images as well as z-stacked widefield microscopy images of individual young or old HSCs (Fig. 2a, b left, one layer shown). For confocal imaging, the medians of $P_n$ of Cdc42 or Tubulin of individual HSCs were elevated in young compared to old HSCs (Fig. 2a right), with a significant difference of the median of $P_n$ of Cdc42 between young and old HSCs. For widefield imaging, the medians of $P_n$ of Cdc42 or Tubulin of individual HSCs were also elevated in young compared to old HSCs (Fig. 2a right), with a significant difference of the median of $P_n$ of Cdc42 and Tubulin between young and old HSCs. Overall, significant differences showed an effect size in the range from a medium to a small effect. These data recapitulate a reduced overall polar distribution of especially Cdc42 in old compared to young HSCs. $P_n$ values of 3D IF images of HSCs (here all z-stacks of the IF images, Fig. 2c) successfully align distinct levels of $P_n$ values with visually scored types of polarity. As semi-automatic acquisition of widefield imaging was established in our laboratory, we focused in all subsequent experiments on widefield IF images.

Cdc42 is a small RhoGTPase[38]. Cdc42 is not only a polarity marker protein, but via its GTPase function, also a regulator of polarity[10,39]. Upon aging, there is elevated activity of Cdc42 in HSCs, which results in the higher frequency of old HSCs being apolar for Cdc42 itself[2,40]. Pharmacological attenuation of the elevated Cdc42 activity in old HSCs to the level of the activity in young HSCs by the drug CASIN (at 5 µM) is known to repolarize old HSCs (by visual determination)[2,17]. We determined therefore the extent of repolarization of old HSCs by CASIN in 3D by CellDetail in comparison to young and old HSCs. Displaying the data as the percentage of HSCs relative to their $P_n$ value for Cdc42 (Fig. 2d) identified that there are more old HSCs with a lower value for $P_n$ and there is a higher percentage of young and old+CASIN HSCs with higher $P_n$ values, implying shifts in polarity between young and aged HSCs along the whole range of values for polarity, while $P_n$ distribution after re-polarization of old HSCs by CASIN aligns more to young HSCs along the range of $P_n$. Second, to be able to compare polarity identified by CellDetail to previously published binary polarity

data on young, old and old HSC rejuvenated by CASIN, we calculated the cumulative percentage of cells along $P_n$ values (Fig. 2e) of the data presented in Fig. 2d. We previously reported that on average 60–65% of young HSCs are polar for Cdc42 distribution (visual examinations,[2]). Using this frequency as the threshold for a binary scoring of polarity like in visual examinations in our analyses (black vertical line in Fig. 2e, which falls onto a $P_n$ of about $0.6 \times 10^{-6}$, which is also the level of $P_n$ that separated polar from non-polar HSCs in Fig. 1h) 49% of old HSCs are listed as polar, while now 68% of old HSCs treated with CASIN are listed as polarized. These frequencies of polar cells of rejuvenated HSCs are in accordance with published frequencies of rejuvenation, while the overall frequency of polar cells among old HSCs, while still significantly reduced compared to young HSCs, is somewhat higher than previously identified by visual examination[2]. The data further show that CASIN treated aged HSCs follow closely the cumulative frequency curve of young HSCs along the range of $P_n$ values, which again implies that all types/levels/states of polarity of aged HSCs respond with an increase in polarization upon CASIN treatment.

In HSCs, nuclei occupy a large volume of the cell (Fig. 2a, f). It therefore remains a possibility that polarity of proteins in HSCs is simply a consequence of the position of the cytoplasmic space driven by the position of the nucleus. To address this question, we determined the correlation between a parameter that identifies the position of the nucleus ($d_{nucleus}$) and the level of protein polarity in the cytoplasm (Fig. 2f, g). The position of the nucleus within the cell was calculated based on the negative charge-weighted center $\mathbf{R}_-$ of the DAPI channel in the cytoplasm (Fig. 2f), as the position of the positive DAPI charge center within the nucleus is strongly affected by, due to their difference in DAPI intensity, the distribution of heterochromatin and euchromatin, and will thus deviate from the real barycenter of the nucleus. The parameter $d_{nucleus}$ was calculated as percentage of the maximal diameter $d_{max}$ (orange arrow): $d_{nucleus} = |\mathbf{MR}_-| / d_{max} \cdot 100$, with $|\mathbf{MR}_-|$ (green arrow) being the distance between $\mathbf{M}$ (the middle of the cell) and $\mathbf{R}_-$ in the DAPI channel (Fig. 2f). We expected a linear relationship of polarity with position of the nucleus and thus used Pearson correlation for calculations ($R_{Pearson}$). There were intermediate strong correlation coefficients of $P_n$ of DAPI and $d_{nucleus}$ ($R_{young} = 0.53$, $R_{old} = 0.64$), as the position of the nucleus and cytoplasm are indeed related to each other in HSCs (Fig. 2g). The low $R_{Pearson}$ values for polarity of Tubulin and nucleus positioning ($R_{young} = 0.13$, $R_{old} = 0.08$) imply that the position of the nucleus within the cell does not simply determine in general the level of polarity of a polarity protein like Tubulin. The intermediate levels of correlation of polarity of Cdc42 and position of the nucleus ($R_{young} = 0.44$, $R_{old} = 0.59$) though could imply a role of the position of Cdc42 in positioning of the nucleus also in HSCs, as has been previously implied in other cell types[41–43].

CellDetail can be readily applied to other types of quantifications of cellular components. We previously investigated changes in the spatial distribution of chromosomes upon aging[44]. Quantification of changes in the position of chromosomes remained challenging, and we finally used homolog distance as an indicator for changes in chromosome distribution (Supplementary Information A, Fig. S1a). Re-analysis of the original image data with CellDetail allows for a quantification of changes of the position of chromosome 11 homologs in the nucleus. CellDetail quantified the distinct distribution of chromosome 11 in the nucleus of young or aged HSCs, and confirmed that the attenuation of Cdc42 activity with CASIN restored a youthful distribution of chromosome 11 in the nucleus of chronologically aged murine HSCs (Supplementary Information A, Fig. S1b).

### The network of the cytoskeletal Septin proteins is less organized in old HSCs

How changes in Cdc42 activity affect polarity within HSCs is not known. We recently demonstrated that the level of activity of Cdc42 determines the localization/polarity of Septin7 within HSCs[31]. Septins

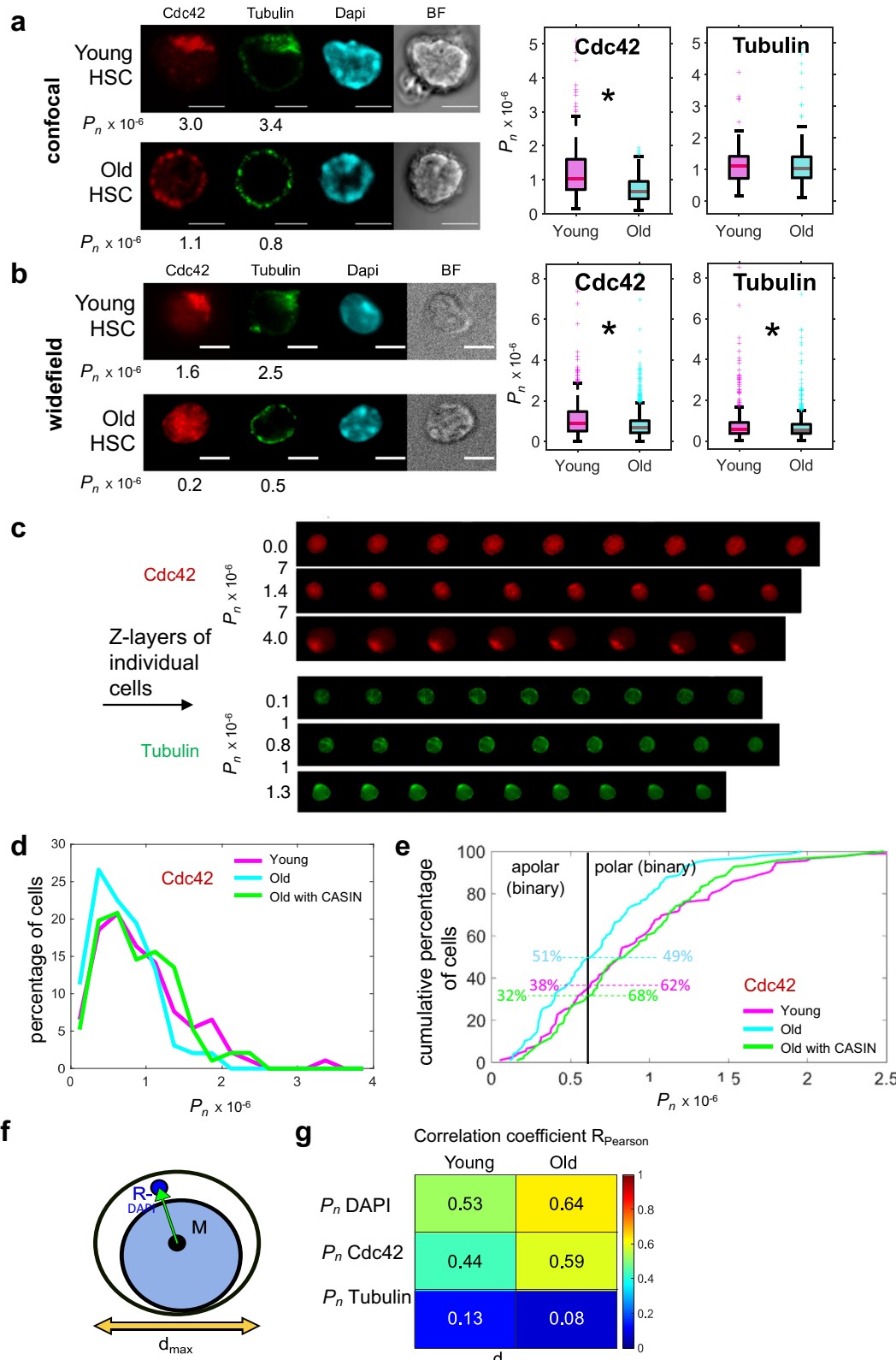

interact with Actin and Tubulin. They compartmentalize cells by forming hetero-oligomeric complexes and filaments which serve as scaffolds for protein recruitment or as diffusion barriers[30,33,45]. There are 13 Septins in mammals, which are classified into four SEPT groups based on homology.

We hypothesized that the localization or distribution of Septins and their network structure in old HSCs is changed, not only for Septin7. A quantitative analysis of changes in the organization of a complex protein-protein network in adult stem cells has been so far elusive due to lack of proper image analysis tools. We determined here the distribution of Septin1, Septin2, Septin6, Septin7, Septin9 and Septin11 in individual HSCs with a newly developed multi-color staining panel (Supplementary Information D, Fig. 3a). The median of $P_n$ for each of the Septins was reduced in old compared to young HSCs

**Fig. 2 | Polarity of Cdc42 and Tubulin in HSCs. a** Left: Confocal IF images (example from z-stacks) of Cdc42, Tubulin and DAPI (scale bar 5 μm). Right: Absolute value of $P_n$, with median as central mark, edges indicating 25th and 75th percentiles, whiskers extending to most extreme data points not considered outliers, outliers individually plotted. Median Cdc42: $P_{n,y} = 1.0e\text{-}6$, $P_{n,o} = 0.7e\text{-}6$, $p = 2.3e\text{-}24$, common language effect size (CLES) $D_{Cdc42} = 0.25$, median Tubulin: $P_{n,y} = 1.1e\text{-}6$, $P_{n,o} = 1.0e\text{-}6$, $p = 0.8$, CLES $D_{Tubulin} = 0.49$, $n_y = 217$, $n_o = 372$ **b** Left: Widefield IF images (example from z-stacks) of Cdc42, Tubulin and DAPI, scale bar 5 μm. Right: Absolute value of $P_n$, with median as central mark, edges indicating 25th and 75th percentiles, whiskers extending to most extreme data points not considered outliers, outliers individually plotted. Median Cdc42: $P_{n,y} = 0.89e\text{-}6$, $P_{n,o} = 0.68e\text{-}6$, $p = 1e\text{-}12$, CLES $D_{Cdc42} = 0.40$, median Tubulin: $P_{n,y} = 0.60e\text{-}6$, $P_{n,o} = 0.55e\text{-}6$, $p = 0.009$, CLES $D_{Tubulin} = 0.46$, $n_y = 630$, $n_o = 1370$. **c** Z-layers of individual cells (3D) and their $P_n$ values of Cdc42 and Tubulin. **d** Percentage of cells over their $P_n$ for young, old, and old HSCs treated with CASIN ($n_{young} = 92$, $n_{old} = 98$, $n_{oldCASIN} = 96$, median $P_{n,young} = 0.81e\text{-}6$, $P_{n,old} = 0.64e\text{-}6$, $P_{n,oldCASIN} = 0.84e\text{-}6$, $p_{young,old} = 0.0032$, $p_{young,oldCASIN} = 8.0e\text{-}4$, $p_{young,oldCASIN} = 0.9$, CLES $D_{young,old} = 0.38$, $D_{old,old\ treated} = 0.36$) (**a–d**) all two-sided Wilcoxon-Ranksum test. **e** Cumulative percentage of HSCs in dependence on $P_n$ value distribution (data from (**d**)). **f** Scheme for determination of $d_{nucleus}$ via $|\mathbf{MR_-}|$ (green arrow) and the maximal diameter $d_{max}$ (orange arrow). **g** Pearson correlation coefficients $R_{Pearson}$ of the $P_n$ of DAPI, Cdc42 and Tubulin vs. $d_{nucleus}$, with $p < 0.05$ for all correlations shown (two-sided). $p_{old,dnucleus,Cdc42} = 2.8e\text{-}129$, $p_{old,dnucleus,Tubulin} = 0.0024$, $p_{old,dnucleus,DAPI} = 7.3e\text{-}157$; $p_{young,dnucleus,Cdc42} = 1.0e\text{-}30$, $p_{young,dnucleus,Tubulin} = 0.001$, $p_{young,dnucleus,DAPI} = 2.0e\text{-}46$. Source data are provided as a Source Data file.

(Fig. 3b), except for Septin 1, with effect sizes in the small to medium range. Septins showed a more apolar distribution in aged HSCs, which implies additional changes in the overall structure of the network. We therefore determined the positions of Septin proteins relative to each other in young and aged HSCs. To this end (i) the distances between positive charge-weighted protein centers $\mathbf{R_+}$ (Fig. 3c) as well as (ii) the angles between the positive charge-weighted centers $\mathbf{R_+}$ (by determining the angle between the lines of the middle of the cell $\mathbf{M}$ to the individual positive protein charge centers) were calculated (Fig. 3e). The distance provides a measure of the similarity between distributions (Fig. 3d). A small distance can be due to two polar distributions close together or due to two apolar distributions, while a large distance is the result of two polar distributions opposite each other in the cell volume. Whether changes in these distances are associated with large changes of the overall location of proteins or with more loose interconnections between proteins was determined via analysis of the angle value. Small angle differences with a single peak indicate more loose inter-connections, while large angle differences with multimodal distributions indicate larger units changes (Fig. 3f). The angle analysis is thus critical for further classifying inter-protein relationships. Aging affected the median distances between the positive charge centers of Septin7 and Septin6 (increase) and between the positive charge centers of Septin7 and Septin9 and between Septin7 and Septin11 (decrease). Thus, the distributions of Septin7 and Septin6 become less similar upon aging, while Septin7-Septin9 and Septin7-Septin11 distributions become more similar. Comparing the angles between positive charge centers of Septins showed an overall significant increase of the median angle for any combination of Septins upon aging (angles between Septin 6 and Septin 7 and between Septin 6 and Septin 11 are shown in Fig. 3e). These data imply that upon aging, there is indeed reduced inter-connectivity among Septins in HSCs.

For another type of determination of the extent of changes in the Septin network in HSCs upon aging, we calculated Pearson correlation coefficients ($R_{Pearson}$) for intensity values for every pair of intensity channel pair combination of the Septins analyzed (implemented in CellDetail). $R_{Pearson}$ describes in this case the amount of co-localization of two Septins with $R = 1$ being the perfect match, $R = 0$ being random and $R = -1$ being the perfect mismatch. Overall, distinct Septin proteins showed significant, but minor changes between young and old HSCs (Fig. 3g), with interactions that include Septin2 or Septin7 being frequently affected by aging. These findings imply that the overall composition of the Septin filaments with respect to individual Septins and their position within the network is indeed altered upon aging, while there is no major reorganization of the composition and order of the basic filament network units upon aging.

## Cdc42 activity determines the organization of the Septin network in HSCs

We also determined the extent to which attenuation of Cdc42 activity in old HSCs by CASIN results, besides in repolarization of Cdc42 (Fig. 2d, e), also in repolarization of Septins and whether exposure to CASIN affects the Septin overall network structure (Fig. 4a). Old HSCs treated with CASIN presented with elevated median values of $P_n$ for Cdc42 itself, Septin6, Septin7 and Septin9 (Fig. 4b) compared to old HSCs. More importantly, the elevated levels of Spearman correlation values $R_{Spearman}$ (due to positive correlations expected, but not necessarily linear ones) for the polarity of Septin6 and Septin7 with respect to polarization of Cdc42 strongly imply that rejuvenated old HSCs (aka CASIN treated HSCs,[2,17]) restore also in part their Septin network (Fig. 4c). Interestingly, the correlation of Septin9 polarity with Cdc42 polarity was not affected by the treatment though its polarity was increased. This might imply a special role for Septin9 in the Septin network. Whether Cdc42 attenuation directly affects the distribution of all Septins, or the distribution of a single Septin like Septin7 that affect the polarity of other Septins will require further investigations.

## Changes in the organization of the Septin network in aged human fibroblasts

CellDetail has been developed for highly isotropic cells like HSCs, but allows analysis of other cell types, including fibroblastoid cells or in general cell line cells. Whether there are changes in the Septin network in senescent cells is not well known. We investigated the effect of aging on the distribution of the Septin network in FF95 human-derived primary fibroblasts to test whether aging also affects this Septin network, and if so, whether there are any similarities to changes of the network in murine HSCs. FF95 cells reach replicative senescence at cumulative population doublings (CPD) above 42 and are considered aged. FF95 cells with CPD 4.3 and CPD 59.0 were stained for Septin6, Septin7, Septin9, Septin11 and Tubulin. Overall, Septin6, Septin7, Septin9 and Septin11 were more clustered around the nucleus while Tubulin was distributed over the total cell body (Fig. 5a). $P_n$ was significantly decreased for Septin6 and for Septin11 in senescent FF95 cells, but interestingly not for Septins7 and 9 and Tubulin (Fig. 5b). For determining the relationships of co-localizations of distinct Septins within individual cells, we first generated a Pearson correlation matrix of colocalizations among Septins and Tubulin for control and senescent FF95 cells, and then calculated the Pearson correlations of the Pearson correlation coefficients, which allows to investigate how the colocalizations among Septins correlate with each other and thus provides a broader view on the Septin network in control and senescent cells (Fig. 5c). The generally high correlation values for both control FF95 and senescent FF95 imply indeed a tight Septin network structure in both types of cells, with some important changes though upon senescence.

There is a strong decrease in the correlation of Septin7-Septin9 to Septin7-Tubulin upon senescence (Fig. 5c). There is further a decrease in correlations of Septin7-Septin9 to Septin6/Septin9/Septin11-Tubulin, and to a smaller, but still prominent extent also for correlations of Tubulin linked to Septin6-Septin9 and Septin9-Septin11 interactions. A further decrease is visible in correlations of Septin7-Septin9 to Septin6-Septin11 or Septin7-Septin11. Thus, in non-senescent cells, proteins are

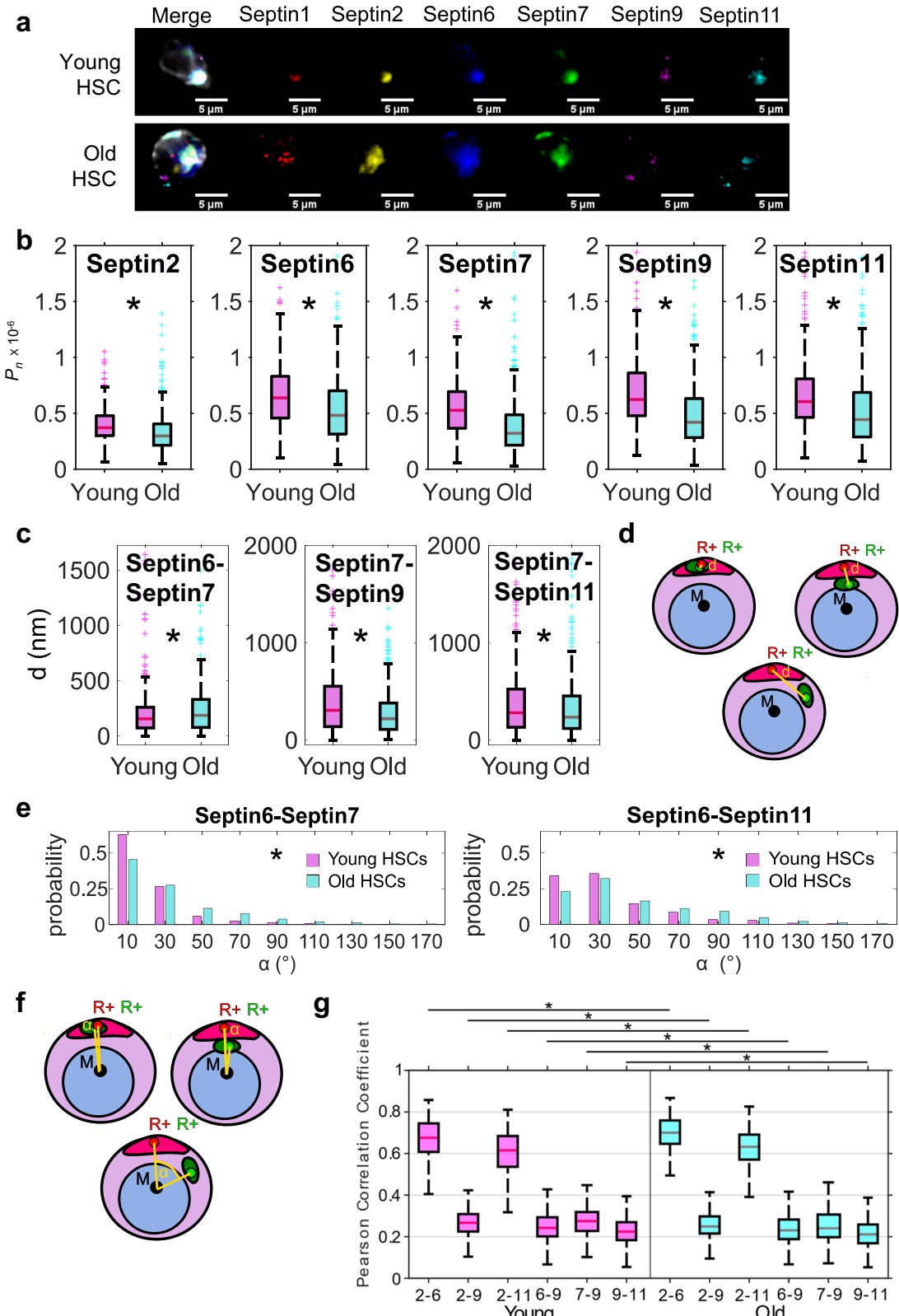

more present in the same region of the cell, while there is a change in the spatial appearance of especially Septin7 and Septin9 upon senescence, as correlations that contain Septin9 show more randomness in combination with other Pearson correlation coefficients, again, implying a special role of Septin9 upon aging/senescence. Interestingly, Septin9 has been implied to play a role in the senescence-associated secretory phenotype (SASP) of senescent fibroblasts via its promotion of secretion of matrix metalloproteinase[32,46,47]. Our results are consistent with the possibility that changes in the position of Septins might contribute to the SASP phenotype of senescent fibroblasts. Thus, and at least in part similar to observations in aged HSCs, the Septin network is more disorganized in senescent human fibroblasts, while it likely maintains its basic composition of type of Septins that form the network.

**Fig. 3 | The organization of the Septin network in HSCs changes upon aging. a** IF images (example from z-stacks) of a young or old HSC (scale bar 5 μm) of Septins 1,2,6,7,9,11. **b** Absolute value of $P_n$, with median as central mark, edges indicating 25th and 75th percentiles, whiskers extending to most extreme data points not considered outliers, outliers individually plotted. Common language effect size (CLES) $D_{Sept2} = 0.35$, $D_{Sept6} = 0.36$, $D_{Sept7} = 0.28$, $D_{Sept9} = 0.30$, $D_{Sept11} = 0.34$; $p_{Sept1} = 0.27$, $p_{Sept2} = 2.0e\text{-}12$, $p_{Sept6} = 5.6e\text{-}11$, $p_{Sept7} = 1.5e\text{-}24$, $p_{Sept9} = 7.1e\text{-}22$, $p_{Sept11} = 1.2e\text{-}13$, $n_y = 330$ and $n_o = 440$. **c** Distance $d$ |between positive charge centers (box plots, median as central mark, edges indicating 25th and 75th percentiles, whiskers extending to most extreme data points not considered outliers, outliers individually plotted), $n_y = 330$ and $n_o = 440$. CLES $D_{Sept6\text{-}Sept7} = 0.45$, $D_{Sept7\text{-}Sept9} = 0.41$, $D_{Sept7\text{-}Sept11} = 0.45$. $p_{Sept6\text{-}Sept7} = 0.03$, $p_{Sept7\text{-}Sept9} = 2.9e\text{-}5$, $p_{Sept7\text{-}Sept11} = 0.03$. **d** Graphical representations of examples of the distance $d$ between

charge centers. **e** Angle $\alpha$ between different positive charge centers of distinct Septins in young or old HSCs, median, $n_y = 330$ and $n_o = 440$. CLES $D_{Sept6\text{-}Sept7} = 0.38$, $D_{Sept6\text{-}Sept11} = 0.40$, $p_{Sept6\text{-}Sept7} = 6.3e\text{-}9$, $p_{Sept6\text{-}Sept11} = 1.4e\text{-}6$. **f** Graphical representations of examples of the angle $\alpha$ between charge centers. **g** Pearson correlation coefficient values ($R_{Pearson,image}$, output parameter of CellDetail) of the intensity channels for distinct combinations of Septins (box plots, median as central mark, edges indicating 25th and 75th percentiles, whiskers extending to most extreme data points not considered outliers, outliers individually plotted), $p_{Sept2\text{-}Sept6} = 2.2e\text{-}4$, $p_{Sept2\text{-}Sept9} = 0.02$, $p_{Sept2\text{-}Sept11} = 0.02$, $p_{Sept6\text{-}Sept9} = 0.01$, $p_{Sept7\text{-}Sept9} = 7.2e\text{-}6$, $p_{Sept9\text{-}Sept11} = 0.006$, $n_y = 330$ and $n_o = 440$. CLES for distributions with significant different medians are: $D_{2\text{-}6} = 0.42$, $D_{2\text{-}9} = 0.45$, $D_{2\text{-}11} = 0.45$, $D_{6\text{-}9} = 0.45$, $D_{7\text{-}9} = 0.41$, $D_{9\text{-}11} = 0.44$. **b, c, e, g** two-sided Wilcoxon Ranksum test. Source data are provided as a Source Data file.

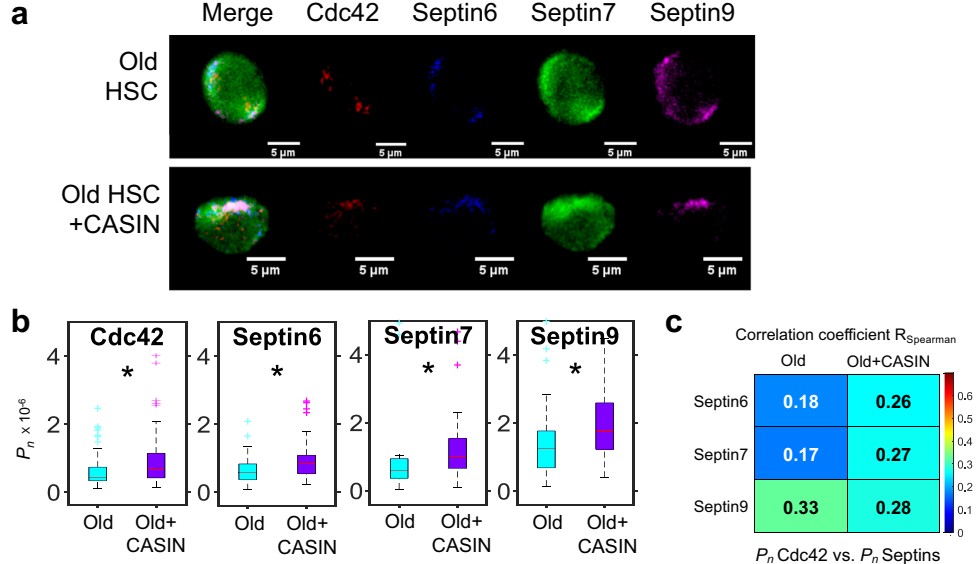

**Fig. 4 | The level of organization of the network of Septin proteins in aged HSCs can be restored to a youthful level by pharmacological attenuation of the activity of the small RhoGTPase Cdc42. a** IF images (examples) of one z-stack plane of old or old HSC treated with CASIN (scale bar 5 μm) of Cdc42 and Septins 6,7,9. **b** $P_n$ of Cdc42 and Septins 6,7, 9 (box plots, with median as central mark, bottom and top edges of box indicating 25th and 75th percentiles and whiskers extending to most extreme data points not considered outliers, and outliers

individually plotted). $n_{old} = 57$, $n_{old,CASIN} = 60$ HSCs the same old mice. Median, two-sided Wilcoxon-Ranksum test, $p_{Cdc42} = 0.01$, $p_{Sept6} = 0.0036$, $p_{Sept7} = 1.5e\text{-}4$, $p_{Sept9} = 3.7e\text{-}4$. Common language effect size (CLES): $D_{Cdc42} = 0.36$, $D_{Septin6} = 0.34$, $D_{Septin7} = 0.30$, $D_{Septin9} = 0.31$. **c** $R_{Spearman}$ of $P_n$ of Cdc42 and $P_n$ of Septins of old and old+CASIN HSCs. $n_{old} = 57$, $n_{old,CASIN} = 60$. $p_{old,Cdc42Sept6} = 0.22$, $p_{old,Cdc42Sept7} = 0.19$, $p_{old,Cdc42Sept9} = 0.01$, $p_{oldCASIN,Cdc42Sept6} = 0.048$, $p_{oldCASIN,Cdc42Sept7} = 0.03$, $p_{oldCASIN,Cdc42Sept9} = 0.03$. Source data are provided as a Source Data file.

## Discussion

Due to the highly isotropic nature of HSCs, the quantitative analysis of protein distribution within HSCs remains challenging. We show that CellDetail is a versatile tool for quantitative analyses of spatial distributions of components in both isotropic cells like HSCs as well as in anisotropic cells like fibroblasts. CellDetail can be especially an asset in research questions concerning spatial behavior of component of interest when the component does not have a spatial correlation with a morphological feature of a cell. CellDetail requires single cell images as input. CellDetail offers the advantage of providing a continuous parameter of the dipole moment ($P_n$) that can be directly analyzed without the need of additional models/tools to generalize data for interpretation[29]. CellDetail works with both confocal and conventional widefield microscopy data. Widefield microscopy analyses have the advantage of less photobleaching. Mid-throughput automated widefield imaging systems are also more readily available and accessible, which will allow obtaining distribution information based on a large number of individual cells to further increase the validity of the data. The GUI makes CellDetail available to the community and allows for usage of CellDetail by trained users without the need of additional programming.

CellDetail does not make use of subcellular segmentation approaches. The information of the whole cell per channel is used for generation of values for spatial distribution. We believe that this might avoid inconsistencies linked to manual protein segmentation or segmentation by neural networks. Lack of segmentation also enables fast automatization as no neural network needs to be trained and no manual adaptation needs to be made. CellDetail also offers more information content compared to a plain barycenter calculation based on calculation over all voxels, as in CellDetail both positive and negative charge center are calculated instead of just one barycenter and spatial distribution information of both low and high component areas can be calculated. Additional parameters determined via CellDetail, like angles between positive charge centers, offer the possibility to quantify the distribution of components in relation to each other. Furthermore, CellDetail offers the possibility to compare spatial distributions independent of the amount of the individual components or signal strength thanks to the average intensity normalization option within CellDetail. It will support, more broadly, the analysis of the overall change in the 3D organization of cells upon aging and other conditions like disease.

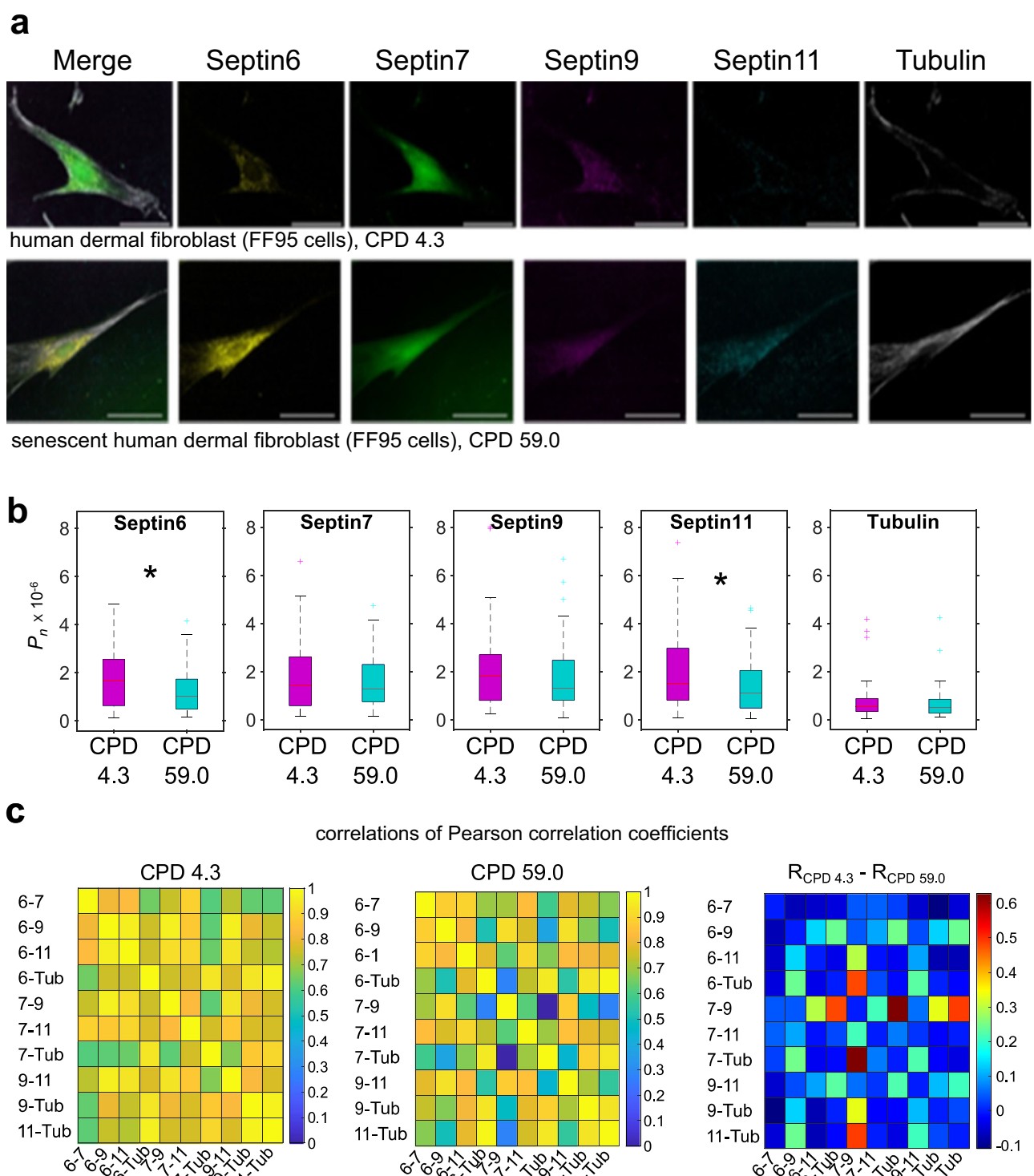

**Fig. 5 | Polarity of Septins and the organization of the Septin network in human dermal fibroblasts is affected by senescence/aging. a** IF images (examples) of one z-stack plane of young and senescent human FF95 dermal fibroblasts for Septins 6,7,11 and Tubulin (scale bar 50 μm). **b** Absolute value of $P_n$, with median as central mark, edges indicating 25th and 75th percentiles, whiskers extending to most extreme data points not considered outliers, outliers individually plotted. Common language effect size (CLES): $D_{Sept6} = 0.39$, $D_{Sept11} = 0.39$, two-sided Wilcoxon-Rank-sum test ($p_{Sept6} = 0.04$, $p_{Sept7} = 0.96$, $p_{Sept9} = 0.4$, $p_{Sept11} = 0.05$, $p_{Tubulin} = 0.6$),

$n_{CPD4.3} = 50$, $n_{CPD59.0} = 68$. **c** Pearson correlation coefficients ($R_{Pearson}$) of paired-channel-based Pearson correlation coefficients ($R_{Pearson}$, output parameter of Cell-Detail) for distinct combinations of Septins and Tubulin of young and senescent human FF95 dermal fibroblasts ($n_{CPD4.3} = 50$, $n_{CPD59.0} = 68$) and right, their difference in correlation coefficients. Except for 7-Tub with 7-9 in CPD 59.0 FF95 cells, all correlations were significant ($p < 0.05$, two-sided; provided within Source Data file). Source data are provided as a Source Data file.

It might be best to initially select just a few polar and apolar cells to determine the type and range of the polarity and how well CellDetail separates them. This might help to inform on which normalization optimization to choose if the standard normalization settings do not apply. Depending on the variance of polarity among cells and difference between populations to compare, CellDetail can work already with as few as 15–30 cells to still provide reliable results, but of course, more individual cells will strengthen the analysis. The numeric output value of CellDetail, the absolute value of normalized dipole moment $P_n$, remains in itself first an arbitrary order of magnitude. This order of magnitude can also shift slightly, depending on which normalization method was chosen. On the other hand, with the normalization average to 1, as chosen for the data analysis in this manuscript, the numeric data output was quite robust among distinct types of analyses. While there will remain a transition zone for $P_n$ values in which we would list cells positive or negative for being polar based on visual examination, a transition as identified in Fig. 2e at around a $P_n$ of $0.6 \times 10^{-6}$ seems to be a good initial benchmark for calling the distribution polar or apolar, which could even be applied to the distribution of chromosome in the nucleus (Supplementary Information A).

CellDetail is limited to the analysis of the distribution of components in single cells. As for all IF experiments, the same staining protocols and measurement settings are recommended to allow for a better direct comparison of data and conditions. Low contrast images might lead to underestimation of polarity, which might be adjusted for by choosing the normalized distance (the parameter is called NormalizedDistanceMaxDiameter or $R_{n1}$) instead of the normalized dipole moment for polarity comparison. For the distribution of filamentous proteins like Tubulin that form climbing-structure on playgrounds like structure within the cell, distinct levels within the cell might show distinct levels of polarity, which in total might though not present as a polar distribution. In such cases, it might be necessary to carefully consider which level of the cell depicts best the biological function of the network in a given setting. The z-resolution can affect the variability of $P_n$. In a setting with a low z-resolution, the z-component of the dipole moment vector is somewhat weaker defined, and $P_n$ might be slightly more variable among cells in comparison to images taken at an isotropic resolution. This might result in a slightly elevated spread of $P_n$ among cells. This effect will be small to negligible in cases in which larger numbers of cells are being analyzed like in our analyses. For example, the spread of $P_n$ values is very similar for the widefield and the confocal data of Cdc42 and Tubulin (Fig. 2a, b), while the underlying images that data is obtained from do indeed vary in axial resolution.

CellDetail allowed us to test whether there is a change in the Septin network structure within HSCs upon aging. The overall position of individual Septins as well as the structure of the Septin network were distinct in young and aged HSCs. We could further demonstrate that the position of distinct types of Septins, and thus likely the whole network structure in aged HSCs, is due to the elevated Cdc42 activity in aged HSCs. The Septin network might be a critical downstream cytoskeletal target in aging and rejuvenation of stem cells via attenuation of Cdc42 activity by CASIN. We obtained comparable results on changes of the Septin network in senescent human FF95 fibroblasts, with some of the Septins being less polar distributed upon aging. The overall basic composition of the network (aka the core composition of the Septin network) with respect to types of Septins though remains not affected by aging, both in murine HSCs and human fibroblasts. Further experiments will need to identify the additional molecular and cellular consequences of a more diffuse Septin network in both HSCs and human fibroblasts. In summary, CellDetail provides complex 3D cell image analyses which allow for a quantitative determination of the spatial distribution of components (e.g., proteins) and component networks in single cells.

## Methods

Our research complies with all relevant ethical regulations and was performed in compliance with German Law for Welfare of Laboratory Animals and approved by the Regierungspräsidium Tübingen. The corresponding protocol number is O.165-5.

### CellDetail

CellDetail was used to analyze IF data. CellDetail is a standalone application based on Matlab code and works on Windows and MacOS. CellDetail needs Matlab Runtime (free of cost), which is downloaded automatically for the web installer files found on github. A manual to CellDetail can be found in the Supplementary information file E Manual of CellDetail. The video tutorial is accessible at https://github.com/xyq91/CellDetail-TS, the program files can be downloaded at https://github.com/xyq91/CellDetail-TS. For the data presented in this manuscript, the intensity of the cell was normalized to average voxel intensity being set to 1 with the charge normalization option $q_{n1}$ (bit depth dependent, cell size dependent) and the distance normalization option $R_{n1}$ (maximal diameter) (Supplementary Information F, Fig. S15).

### Validation data

For code validation an artificial sphere of 5000 nm radius was taken as the object and charges were placed at distinct positions. Different scenarios were calculated both analytically and by usage of the algorithm. The results were in agreement. The scenarios and the comparison of analytical and algorithm solution can be found in the Supplementary Information J (Validation of code - scenarios), a single example scenario can be found in Supplementary Information G (Validation of code - example scenario).

### Mice

Female C57BL/6JRj mice (10–18 weeks old) were purchased from Janvier (St. Berthevin Cedex, France). Aged female C57BL/6J mice (>80 weeks) were obtained from internal divisional stock (derived from mice obtained from Janvier). All mice were housed in Ulm University animal barrier facility under pathogen free conditions and with dark/light cycle 12/12, 22 °C and 60% humidity. All experiments were performed in compliance with German Law for Welfare of Laboratory Animals and were approved by the Regierungspräsidium Tübingen.

### Isolation of LT-HSCs

After isolation of bone marrow from femur and tibia, mononuclear cells were isolated by low-density centrifugation (Ficoll-Paque PREMIUM 1.084, Sigma- Aldrich). The mononuclear cells were stained with a cocktail of biotinylated lineage antibodies: anti-CD5 Biotin (eBioscience, Clone 53-7.3), anti-B220 Biotin (eBioscience, Clone 53-7.3), anti-CD11b Biotin (eBioscience, Clone M1/70), anti-CD8a Biotin (eBioscience, Clone 53-6.7), anti-Gr-1 Biotin (eBioscience, Clone RB6-8C5), anti-Terr-119 Biotin (eBioscience, Clone TER-119). Dynabeads (Sheep anti-rat IgG Life Technologies/Invitrogen) were used for magnetic depletion of lineage cells. Remaining cells were stained with anti-SA-e450 (eBioscience, Streptavidin eFluor 450), anti-Sca-1-PeCy7 (eBioscience, anti-Mo Ly-6A/E, Clone D7), anti-ckit-APC (eBioscience, Clone ACK2), anti-Flt3-PE (eBioscience, Clone A2F10) and anti-CD34-FITC (eBioscience, Clone RAM34). LT-HSCs were gated as lineage negative Sca-1 positive C-Kit-positive (LSK), CD34 negative and Flt3 negative cells[31], Supplementary Information E.

### FF95 cells

Primary human dermal fibroblasts (HDF) FF95 were previously established from foreskin of a one- year-old healthy male having undergone circumcision as described in refs. 48,49. FF95 were passaged to reach their senescence at cumulative population doublings (CPDs) of 59.0, while the non-senescent control cells were passaged less than 5 times.

## IF staining and measurement

Freshly sorted LT-HSCs were seeded on fibronectin-coated glass coverslips (RetroNectin, Takara Bio) in HBSS (Lonza) with 10% FCS (FCS, Sigma- Aldrich, Batch BCCC8748) and 1% Pen/Strep (PAN Biotech). For CASIN treatment, cells were incubated for 14 h in 5 μM CASIN in IMDM in the presence of SCF, G-CSF and TPO before fixation. For confocal imaging data of Cdc42, Tubulin and DAPI, 217 young HSCs (from 7 distinct young samples with 2 young mice pooled per sample) and 372 old HSCs (from 12 distinct old samples with 1 mouse per sample) were analyzed.

For widefield imaging data of Cdc42, Tubulin and DAPI, 630 young HSCs (from 6 distinct young samples with 2 young mice pooled per sample) and 1370 old HSCs (from 10 distinct old samples with 1 mouse per sample) were analyzed. For imaging of young or old HSC for Septins 1,2,6,7,9,11, 330 young and 440 old HSCs were analyzed (HSCs from 6 young mice individually analyzed, HSCs from 8 old mice individually analyzed).

Primary human dermal fibroblast (HDF) strain FF95 cells of cumulative population doubling time (CPD) 4.3 and 59.0 were seeded on fibronectin-coated glass coverslips in DMEM supplemented with L-Glutamine with 10% FCS and 1% Pen/Strep. After two hours LT-HSCs and after 24 h FF95 cells were fixed with Cytofix fixation buffer (BD Cytofix fixation buffer, BD). The cells were washed with PBS, permeabilized with 0.2% Triton X-100 (Sigma-Aldrich) for 20 min and blocked with 10% Donkey Serum (Jackson ImmunoResearch) in 0.1% Triton X-100 for 20 min. For Cdc42 and Tubulin staining, primary antibody staining was performed overnight at 4 °C, secondary antibody staining was performed for 1 h at room temperature. For Septins and Tubulin staining, Tubulin primary antibody staining was performed for 1 h at room temperature before secondary antibody staining together with primary labeled antibody staining was performed for 1 h at room temperature. For Cdc42 and Tubulin staining, the mounting medium consisted of Vectashield with DAPI (Vector Laboratories), for the Septin staining, the mounting medium consisted of Vectashield without DAPI (Vector Laboratories). The Septin staining was planned with fpbase.org spectra viewer.

The antibodies used were (i) anti- Cdc42 (HSC: 1:250) (Abcam, ab64533), anti-Tubulin (HSC: 1:1000, FF95: 1:1000) (Abcam, ab6160), anti-rat AF488 (HSC: 1:500, FF95: 1:2000) (Jackson ImmunoResearch, 712-545-153), anti-rb AF594 (HSC: 1:500) (Jackson ImmunoResearch, 711-585-152), (ii) anti-Septin1-AF790 (HSC: 1:20) (Santa Cruz Biotechnology, sc-373925 AF790), anti-Septin2 (HSC: 1:100) (Abcam, ab58657) labeled with AF405 (Swift Dye Labeling Kit Alexa Fluor 405, G Biosciences, 786-1641-405), anti-Septin6-AF546 (HSC: 1:100; FF95: 1:500) (Santa Cruz Biotechnology, sc-514781 AF546), anti-Septin7 (HSC: 1:1000; FF95: 1:250) (Proteintech, 13818-1-AP) labeled with Atto647N (Atto647N Protein Labeling Kit, Merck, 76508), anti-Septin9 (HSC: 1:100, FF95: 1:1000) (Proteintech, 10769-1-AP) labeled with Qdot705 (Thermo Scientific, S10454), anti-Septin11 (HSC: 1:5000, FF95: 1:8000) (Proteintech, 14672 1-AP) labeled with Qdot585 (Thermo Scientific, S10451), and again anti-Tubulin (HSC: 1:1000) (Abcam, ab6160), anti-rat AF488 (HSC: 1:1000) (Jackson ImmunoResearch, 712-545-153). For the staining of Cdc42 and Septin6 in the CASIN experiment of Septins within HSCs, anti-Cdc42 (HSC: 1:250) (Abcam, ab41429), anti-mouse AF488 (HSC: 1:500) (Jackson ImmunoResearch, 715-545-150), anti-Septin6 (HSC: 1:100) (Invitrogen, #PA5-19024) and anti-goat AF546 (HSC: 1:500) (Thermo Scientific, #A-11056) were used. Both Septin1 and Septin2 showed in FF95 cells, besides their cytoplasmic, also a strong nuclear presence, which did preclude an analysis of their distribution without further removal of the nuclear signal. This approach was not further pursued as all the other Septins analyzed presented with exclusive cytoplasmic distribution. For all stainings, single color controls were used and staining protocols were optimized for avoiding bleed-through of channels (Supplementary Information D).

Samples were either measured with LSM710 confocal microscope (Zeiss) equipped with a 63x water objective (i) or with Axio Scan.Z1 widefield microscope (Zeiss) equipped with a 40x air objective (i, ii). For confocal microscopy, single cells were measured whereas for widefield microscopy, a sample volume was measured and cells were identified for single cell analyses afterwards. For the 7 color IF measurements, the filters Semrock FF01-595/31, Semrock FF01-676/29, Zeiss PBP 425/30 + 514/31 + 592/25 + 681/45 + 785/38, Zeiss QBP 425/30 + 524/51 + 634/38 + 785/38 and Zeiss TBP 467/24 + 555/25 + 687/145 were used with the beam splitters TBS 450 + 538 + 610, QBS 405 + 493 + 611 + 762, PBS 405 + 493 + 575 + 654 + 761, BS573, BS652 and with Zeiss Colibri 7 LEDs (385 nm LED, 430 nm LED, 475 nm LED, 555 nm LED, 590 nm LED, 630 nm LED, 735 nm LED). Stacks of.tiff images of individual cells were processed with CellDetail.

## Statistics and reproducibility

Two-sided Wilcoxon-Ranksum test, which is equivalent to Mann–Whitney U- Test, was used for comparing parameter medians of samples for statistical significant differences by using Matlab's ranksum function. The common language effect size was calculated by determining the Mann–Whitney U statistic, which is the smaller of $U_1 = n_1 \cdot n_2 + n_1 \cdot (n_1 + 1) / 2 - R_1$, $U_2 = n_1 \cdot n_2 + + n_2 \cdot (n_2 + 1) / 2 - R_2$ with $n_1$ being the size of sample 1 and $n_2$ being the size of the sample 2, and $R_1$, respectively $R_2$ being the sum of ranks in group 1, respectively group 2. The area under the receiver operating characteristic curve is the common language effect size, which is $f = min(U1, U2)/(n1 \cdot n2)$. Two-sided Pearson correlation coefficients were calculated by using Matlab's corrcoef function and for the Pearson correlation coefficient of channel overlap a custom code was written following Pearson correlation coefficient calculation based on images (implemented in CellDetail). Two-sided Spearman correlation coefficients were calculated by using Matlab's corr function with specification of type for Spearman. No statistical method was used to predetermine sample size.

## Reporting summary

Further information on research design is available in the Nature Portfolio Reporting Summary linked to this article.

## Data availability

All relevant data supporting the key findings of this study are available within the article and its Supplementary Information files. The data generated in this study are provided in the Source Data file. Source data are provided with this paper.

## Code availability

CellDetail has a graphical user interface (GUI) and is available for download under https://github.com/xyq91/CellDetail-TS (https://doi.org/10.5281/zenodo.13860319)[50]

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

## Acknowledgements

We thank the Core Facility Cytometry for cell sorting support and the Core Facility Confocal and Multiphoton Microscopy at Ulm University for support with confocal micoscopy, and the Tierforschungszentrum of Ulm University for supporting our animal work. We thank Ingmar Glauche and Thomas Zerjatke of the Institute for Medical Informatics and Biometry, Dresden University of Technology for their input on the manuscript. This

work was supported by grants RTG 1789 CEMMA, FOR 2674 and SFB 1506 (all DFG) to H.G. and SFB 1506 to A.G.. The research of K.S-K. is funded by the DFG within the Collaborative Research Centre (CRC1506; Project-ID 450627322) "Aging at Interfaces" and by the Graduate Training Group GRK 1789 "Cellular and Molecular Mechanisms in Ageing (CEMMA)". (FOR 2674: Aging-related epigenetic remodeling in acute myeloid leukemia, Project identifier Deutsche Forschungsgemeinschaft (DFG)- Project number 336840530).

## Author contributions

T.S. developed algorithm concept. T.S. programmed, designed and validated CellDetail. T.S. and H.G. were involved in experimental planning and interpretation of results. T.S., G.M., J.N., K.S., and M.V. performed experiments. A.G. provided data. T.S. analyzed data. T.S., A.A., J.N., M.S., M.V., A.V., and H.G. discussed the algorithm and software CellDetail. T.S., H.G., K.S.K., and M.W. wrote and edited manuscript. K.S.K. and M.W. provided material. V.S. provided reagents. T.S. performed benchmarking analysis and revision experiments. M.M. and H.G. gave guidance for benchmarking analysis.

## Funding

## Competing interests

The authors declare no competing interests.
