## [Transparent Peer Review file · Nature Communications]

Quantitative determination of the spatial distribution of components in single cells with CellDetail

Corresponding Author: Professor Hartmut Geiger

Version 1:

Reviewer comments:

Reviewer #1

(Remarks to the Author)

The manuscript by Schuster et al. reports a novel cell polarity measurement tool, CellDetail, and its application in analyzing subcellular dynamics of known polarity markers Cdc42, Tubulin, and Septins during cell aging. The authors took inspiration from the electric dipole moment measurements common in Chemistry and demonstrated how this metric can quantify polarity changes during the aging of HSCs. Based on the reported data from the manuscript, the tool seemed sensitive and reliable across experiments. The large data sets (hundreds of cells) analyzed here are impressive.

However, several factors dampen my enthusiasm:

Major comments:

1) The abstract suggests that tools to quantitatively determine the polarity of components inside cells are not well developed, but, in fact, there are more published tools created by developmental biologists for this purpose than the few mentioned in the introduction section (lines 61-68). Some are listed below. It would be good to include a comparison between CellDetail and previously reported polarity metrics in the manuscript. Without such a comparison, it will be difficult for readers to recognize the benefits of CellDetail and, thus, the significance of the work reported here.

One way in which the authors could compare the different tools is to use the polar distribution of Cdc42 and Tubulin to train a binary classification model and see which tool does a better job at distinguishing young and aged HSCs (ROC curves would help). I think such a classifier may even prove beneficial in a diagnostic setting.

<https://doi.org/10.1073/pnas.0906227106> (2009)

<https://doi.org/10.1242/dev.146837> (2017)

<https://doi.org/10.1038/s41567-018-0358-7> (2019)

<https://doi.org/10.1111/nph.17165> (2020)

<https://doi.org/10.1242/dev.198952> (2021)

2) While I find the chemistry-inspired dipole moment as a metric of polarity intriguing and novel, I have reservations about its implementation in image analysis. Unlike atoms that have unique charges, fluorescence intensity measurements are sensitive to limits of detection and signal saturation, and I would like a more robust explanation of how normalizing total intensity to 1 and normalization of charge by bit-based and half-cell size could render the dipole moment insensitive to these challenges.

Overall, the authors should better explain what the nine normalization options in CellDetail are good for (SI lines 49-50) and the kind of artifacts they could generate.

3) In the same vein as above, the authors suggest normalizing the distance between charge centers by maximal diameter to compensate for differences in cell size. However, this normalization would cause the dipole moment to be more sensitive to cell shape. For instance, the maximal diameter of a square is larger than that of a hexagon of the same area, and thus, even if a protein is entirely localized to a single vertex in both cells, the resulting normalized dipole moments will be different. This may not be a problem when analyzing isolated round cells like HSCs, but the application of CellDetail to other cells could be quite limited. In fact, I would like to see a comparison of dipole moment for Cdc42, Tubulin, and Septins versus cell circularity for HSCs to determine if any correlation exists.

4) Unlike other previously published metrics of polarity, the dipole moment is difficult to interpret and unintuitive. That is, the dipole moment values throughout the experiments range from $10e-10$ to $10e-13$, and a higher number does not necessarily mean more polarized localization when comparing two different proteins. For example, it's difficult to judge whether Cdc42 and Tubulin are polarized from the average value of their dipole moments alone. The fact that the average dipole moment for the same protein (and I assume from the same sample) can change by an order of magnitude going from a confocal to a widefield microscope is unsettling. Moreover, Cdc42 and Tubulin are unarguably polarized in young HSCs and not in old HSCs based on the images, but the statistical difference between their dipole moments is rather modest. Here, visual inspection of images seems more informative than the quantifications in Figure 2.a.

I would like to see the calculated dipole moments for the images in Fig.2a. printed above each image, as I wonder how these values compare with the averages shown on the box plots.

5) How three-dimensional information of protein localization in the cell can be incorporated into the polarity measurement of CellDetail is unclear. The authors stated that CellDetail can work with both z-stacked confocal images and widefield images (line 190). However, there are no clear examples of z-stacked image measurement or discussion of how cell orientation and differences in X/Y and Z axial resolutions can affect the dipole moment. In fact, I believe I can see the differences in axial resolution in the supplemental videos, so it would be good to quantify if there is any correlation between the measured dipole moments and the Z axis in these z-stacks (e.g., similar to the analysis presented on Fig.6b).

Minor comments:

6) The aspect ratio of Figure 3 is unsuccessful, and it would be problematic in the PDF version of the manuscript when published. Please consider splitting this figure into two and rebalancing the size of the different panels. For example, the box plots are much smaller than the cartoons of the polarized cells in panels d, e, and i, and it is not easy to see the quantitative differences between dipole moments, which I believe is the key finding of this analysis. Moreover, the line graph in panel c seems unnecessarily big for the information the authors seek to convey with this graph.

7) The methods section should list the filters used to separate the 7-color staining in Figure 5. Also, Figure 5 would be more effective if each channel were shown as a heatmap to reassure readers that the steps taken to prevent bleed-through between channels did not cause the fluorescence in a particular channel to be much lower than the others.

8) The use of box plots for comparing dipole moments may not be the most effective way to display the data. The difference in the average dipole moments between comparison groups in Fig. 2, Fig. 3, and Fig. 4 is modest. On the other hand, the distribution levels seem to be changing more dramatically between young and old HSCs, and density curves or histograms would make the visual comparisons much easier.

9) Please ensure that the corresponding p-value for asterisks in all figures is described on the legend for each figure (e.g., the asterisks in Fig. 2.a.i, Fig. 2.a.ii, Fig. 3.b, etc.)

Reviewer #2

(Remarks to the Author)

I co-reviewed this manuscript with one of the reviewers who provided the listed reports as part of the Nature Communications initiative to facilitate training in peer review and appropriate recognition for co-reviewers.

Reviewer #3

(Remarks to the Author)

The ms presents an algorithm for quantification of polarity in the spatial distribution of molecular concentrations in cell microscopy images. The authors package the algorithm in a self contained computer code with a graphical user interface. The applicability of the algorithm is subsequently demonstrated in the analysis of cytoskeletal protein distributions in hematopoietic stem cells and in fibroblasts.

Over the past decades, classifications of cellular phenotypes have undergone a much-needed shift from the manual qualitative and subjective descriptions to reproducible automated quantifications. As such, the goal of the submitted ms is laudable and useful. However, in my opinion the tool and the underlying analysis presented in the ms fall short of the requirements for innovation, quality and general-interest that are expected from a Nature Communication publication. Essentially, the tool calculates the first moment of the protein concentration distributions. The authors place considerable emphasis on recasting the concentration values by subtracting the cell-averaged concentration, thereby creating regions of "negative" and "positive" concentrations. However, I do not see how this provides any extra information, as claimed by the authors. Moreover, multiple numerical quantification schemes for polarity of protein concentrations have been proposed in utilized in the past: Fourier analysis [Merkel et al 2014, Banerjee et al 2017,...]; PCA [Ee Tan et al 2021]; relating concentration polarity to cell edge feature [Farrell et al 2017]; and many others. The authors mention that other schemes suffer from limitations, but since they do not evaluate the benefits of their simplistic approach relative to other procedures, the reader cannot assess this claim. Finally, I found the paper to be overly verbose and with poor, often unintelligible, use of language.

In conclusion, I do not believe that the presented tool goes further than what is available from multiple available plug-ins for common image processing tools, or the procedures described in the methods sections of numerous publications. The application of the analysis for cytoskeletal protein distributions in HSC and fibroblasts does not provide any significant new results or insights. I therefore suggest that this ms may be more suitable for publication in a specialized journal for image processing tools.

Reviewer #4

(Remarks to the Author)

General comments:

Schuster et al. designed a platform, named CellDetail, to quantify biomolecule distribution in single cells. The algorithm uses the concept of dipole moment, a physical quantity that represents the distribution of positive and negative charges. This paper adapted it to identify regions of high and low protein density by imaging, and regarded them as positive and negative charges respectively. These measurements were then used to quantify the spatial distribution and degree of polarity of protein networks. Authors showed that this algorithm can detect the previously reported polarity change of Cdc42 and Tubulin in HSCs during ageing. They also reported that the disruption of the Septin network, both in aged HSCs and in senescent human fibroblasts which was included as another ageing model to evaluate CellDetail's capability of measuring protein localization dynamics.

Overall, CellDetail seems innovative in its application of physical principles to biological image analysis, offering a tool to evaluate biomolecules in single cells. However, my major concern is that the algorithm is specifically designed to measure the spatial distribution of biomolecules, which could limit its application in a broader scenario. Although experiments were performed to validate the algorithm, they provide limited new biological insights. It would greatly improve the manuscript by providing more evidence that the quantitative value of polarity determined by CellDetail can be translated into biological processes.

Specific comments:

1. The rationale of using a quantitative approach, rather than binary measurement, is that it would be more sensitive in detecting subtle changes of protein distribution. However, the change of Pn between young vs old HSCs seems very minimal (marginal significance and fold change for Cdc42, and not significant for Tubulin with bright field images). This seems inferior to the previously reported several folds of decrease of polarized HSCs during ageing, reported by the same group. Could the authors elaborate on this?
2. It would be helpful to show some examples where CellDetail can detect a subtle change that is not detectable or is challenging by visual assessment. Can authors test and explain in more scenarios and show how does this method outperform binary measurement?
3. Since Pn is a quantitative value, it would be interesting to know if this value has a correlation with other features/functions of HSCs. For example, does a stem cell with high Pn show strong repopulating ability? What about their cycling status, proliferation etc.? Does the Pn value have correlations with the immunophenotype of HSCs (e.g. expression levels of stem cell markers/genes)? If one can demonstrate (at least to some extent) that the Pn value is predictive of HSC functions/phenotypes, it would make this algorithm a more powerful tool to study HSC biology.
4. It is not entirely clear how the value of Pn translates to polar vs apolar states. Is it possible to set a threshold based on Pn and annotate cells as polar vs apolar (vs intermediate)?
5. In their previous work, this group demonstrated a decreased polarity of Cdc42 upon aging and showed that restoring Cdc42 distribution by treating aged HSCs with Casin can also revert the differentiation pattern of aged HSCs to more young-like. It would be helpful to show that this rescue of distribution by Casin can be detected using CellDetail as well.
6. The algorithm reported that the Septin network polarity seems to be altered during ageing. Could this be achieved by visual assessment? A direct comparison of the two methods would be informative.
7. The authors hypothesized that upon ageing, elevated Cdc42 level alters the localization of the Septin network. They determined Septin protein distributions and their relative distances to Tubulin, but not Cdc42. Since the hypothesis is that Cdc42 affects Septin network, it would be more informative to also show the polarity of Cdc42 together with Septins to see if there is a direct correlation between the two.
8. Imaging proteins in single cells is often a difficult task, especially for cells with irregular morphology or proteins with low expression level. How does CellDetail deal with challenging images, e.g. suboptimal density, low image quality etc.?

Reviewer #5

(Remarks to the Author)

Changes in the distribution of intracellular biomolecules have been found in aging and diseases. In this article, the authors developed an open-source image analysis tool, CellDetail, to quantitatively determine the polarity of molecules inside cells in 2D and 3D. CellDetail uses a physics-motivated algorithm to quantify the intracellular polarity of proteins as a real number P_n . This is new and innovative. The authors used CellDetail to analyze 6 individual Septins within a single HSC and discovered the reduction of Septin organization and polarity within aged HSCs, as well as in senescent human fibroblasts. CellDetail provides useful image analysis tools for a quantitative determination of the spatial distribution of protein-protein network in single cells. However, I do have concerns with the effectiveness of the quantification method and interpretation of the analysis of results. So, it may be suitable for publication in Nature Communication after the following concerns are addressed.

Major concerns:

1. Figure 1. How was the center point M calculated? The authors wrote that the program can be downloaded under <https://cloudstore.uniulm.de/s/CnarsbtJ3gYiWww> However, this link does not seem to work.
2. It would be helpful to provide some intuitive understanding of the physical mean of P_n . Figure 2. What are the P_n values corresponding to the young and old HSCs shown in Figs. 2a(i)-(ii)? Are these cells representative?
3. When the nuclear position and CDC42/tubulin polarity was compared, did the authors use confocal image data or widefield image data?
4. Figs 3d and 3e, small distance represents both polar with similar R_+ and apolar. Is there any way to separate these two representations? Apolar showed random angles, including both large angles and small angles. This means that small angles could represent either polar with similar R_+ or apolar. And large angles also could represent either polar with different R_+ or apolar.
5. Figure 2. Pearson correlation coefficients for polarity of Cdc42 and nucleus positioning ($R_{\text{young}} = 0.60$, $R_{\text{old}} = 0.57$), and especially for polarity of Tubulin and nucleus positioning ($R_{\text{young}} = 0.32$, $R_{\text{old}} = 0.16$). Figure 3. The correlation coefficients between P_n and ζ_+ were between 0.2-0.67 considered correlated. What is the physical meaning of Pearson correlation coefficients and is there a uniform measure of what could be interpreted as correlated or not correlated?
6. Figs. 3k and 4d. Pearson correlation coefficient of intensity channels. How were the correlation coefficients and intensity channels computed?
7. Figure 3k, the author stated that "For Septin1-Septin2, Septin1-Septin6, Septin1-Septin7, ... and Septin9-Tubulin significantly different medians were found. Nonetheless, the Pearson correlation coefficients among the Septin proteins and also Tubulin remained similar among young and old HSCs". The statistics show many septin groups are different between young and old HSCs, yet the author concludes that they remained similar. The conclusions are just not convincing. Are the septins significantly different or not?
8. Figs. 4b-4c, did tubulin also show a trend of change? It is not convincing what is the reason that the authors conclude that tubulin did not change upon senescence while some septins showed change. Figs. 4d-4e. There is a strong decrease in the correlation of septin7-septin9 to 7-tub collocation, what is the cause of this? From the results show in Fig. 4d, 7-9 correlation increase upon senescence (although non-significantly). How do we interpret this result?
9. Figure 6a. It seems that p-value needs to be corrected for multiple comparison test.

Version 2:

Reviewer comments:

Reviewer #2

(Remarks to the Author)

I thank the authors for their efforts to address the previous concerns of the reviewers' comments, and I believe the revision has greatly improved the logic and flow of the manuscript. However, I am still not convinced of the sensitivity and reproducibility of CellDetail.

Major comments:

- 1) While the benchmarking of CellDetail using Cdc42 in comparison with some other published polarity quantification tools has suggested that CellDetail can accurately quantify Cdc42 polarity (about the same accuracy as several published toolkits), there is limited evidence to suggest that CellDetail can be readily applied to other cell polarity quantification. Without other test cases, it is difficult to convince readers that CellDetail can be used in their own polarity systems.
- 2) It is concerning that CellDetail failed to detect the difference in tubulin polarity in young and old HSCs from confocal images, even with the sample size of more than 500 cells and being one of the positive controls that the authors selected (Figure 2a). Can such a difference be picked up by other existing polarity quantification tools? Is that a problem of the tool or the hypothesis? The same goes for Septin1 measurements in young and old HSCs. In the first submitted version of the manuscript (Figure 3b), there was a significant difference in the polarity quantification output by CellDetail. In the current revision, such difference however is not present anymore (line 281). As a matter of fact, no quantification of Septin1 polarity is included in the main manuscript or supplementary data anymore, prompting me to doubt the authors' confidence in CellDetail themselves.
- 3) In the same vein as above, I am less impressed by the application of CellDetail in fibroblast polarity quantification. The two Septins (6 and 11) that exhibit different dipole moment quantifications (Figure 4) are clearly expressed at drastically

different levels. Even with all the normalization that the authors included in the CellDetail pipeline, it is still a fair question whether the different readout is because of the low signal in one or the actual polarity difference. I think what is missing here are a positive control to show the sensitivity of CellDetail in less isotropic cells and a direct assessment of how expression level, staining intensity, or signal-to-noise ratio is affecting CellDetail quantification.

4) I greatly appreciate the improvement that the authors made to Figure 1 to make the measurement of dipole moments from images much easier to grasp. However, how positive or negative charges (and the subsequent determination of charge centers) are determined is still not clear. I would suggest the authors add such info to Figure 1e, maybe with masks of positive and negative of charges over the fluorescent image on the side and the pinpoint of the charge centers.

5) Boxplots in all the polarity measurements are insufficient to represent the distribution of the individual measurement. I would suggest showing all the raw measurements or including the distributions of all the data points in all the quantification output plots.

6) No limitation of CellDetail is included in the manuscript. The authors should include a fair assessment of the limitations of the tool in the discussion. Also, there should be a section discussing how to interpret the direct numeric outputs by CellDetail to assess the polarity level.

Reviewer #3

(Remarks to the Author)

In the revised version of the ms the authors have gone a long way to address the concerns that I and other reviewers had raised.

Most importantly, the revised ms now thoroughly compares the suggested approach (first moment of the distribution) to other standard polarity quantification schemes in terms of accuracy and robustness. In this regard, the quality of the work is increased significantly. There is still a lack of a theoretical analysis of this approach, and I maintain that the "dipole moment" terminology is misleading, as there is only one "charge" in the system - but it may be of practical use to the community. My earlier comments regarding the disjointed nature of the septin part stand, however.

Reviewer #4

(Remarks to the Author)

In the revised manuscript, the authors have added benchmarking analysis of CellDetail to compare it with other existing tools, and included additional work to determine and improve their tool's robustness. They also provided more details regarding the methodology behind CellDetail. Using CASIN treatment as a mean to test this tool's ability to detect protein network changes during the rejuvenation of old HSCs, they showed that CellDetail can be used to monitor subtle changes of biomolecule distributions under certain biological process, such as ageing. While it remains unclear if the polarization value (P_n) calculated by CellDetail is linked to specific cell functions due to technical challenges, this revised manuscript has adequately addressed my concerns.

Reviewer #5

(Remarks to the Author)

Thank you for the response. I suggest to accept this paper for publication.

Version 3:

Reviewer comments:

Reviewer #2

(Remarks to the Author)

Thank you for the response. The additional quantification and explanation is helpful for the interpretation of P_n . I suggest to accept the manuscript for publication if the editors feel adequate. However, this reviewer is still skeptical of the sensitivity and reliability of CellDetail and the potential adaption of this tool by the larger cell polarity community.

NCOMMS-23-35503A-Z Point-to-Point response

Dear Reviewers,

we are honored by the fact that all reviewers saw novelty and value in our novel tool (CellDetail) for the analysis of distribution of bio-components within HSCs and cells with other types of morphology like fibroblasts. CellDetail allowed the determination of the changes in the distribution of Septins and the Septin network in aged stem cells and senescent fibroblasts. Septins are the fourth component of the cytoskeletal network, and are major players in regulating compartmentalization of cells. More recently there is growing attention with respect to the role of Septins in biology, as they so far have not been studied in great detail in comparison to other components of the cytoskeletal network.

See below our Point-to-Point response to the comments of the Reviewers.

Reviewer #1:

The manuscript by Schuster et al. reports a novel cell polarity measurement tool, CellDetail, and its application in analyzing subcellular dynamics of known polarity markers Cdc42, Tubulin, and Septins during cell aging.....Based on the reported data from the manuscript, the tool seemed sensitive and reliable across experiments. The large data sets (hundreds of cells) analyzed here are impressive.

Response: We thank the reviewer for this very positive evaluation of the value of CellDetail.

Major comments:

1) The abstract suggests that tools to quantitatively determine the polarity of components inside cells are not well developed, but, in fact, there are more published tools created by developmental biologists for this purpose than the few mentioned in the introduction section (lines 61-68). Some are listed below..... It would be good to include a comparison between CellDetail and previously reported polarity metrics in the manuscript. Without such a comparison, it will be difficult for readers to recognize the benefits of CellDetail and, thus, the significance of the work reported here. One way in which the authors could also compare the different tools is to use the polar distribution of Cdc42 and Tubulin to train a binary classification model and see which tool does a better job at distinguishing young and aged HSCs (ROC curves would help). I think such a classifier may even prove beneficial in a diagnostic setting.

Response: We performed a large set of benchmarking analyses (**Supplementary Information A, pages 3 to 8, Fig.S1, Table S1**). The benchmarking was performed against the existing methods i) barycenter method with/without normalization, ii) intensity profile fit method, iii) intensity ratio method, iv) Fourier method, v) PCA method, vi) neural network, and used 2D layers of cells visually selected for polarity and apolarity for Cdc42 as the images to be analyzed. As most of the other tools work only on/in 2D, a truly comparative benchmarking analysis could only be performed on 2D images. CellDetail was more accurate in separating image layers of HSCs visually scored for being polar or apolar for Cdc42 in comparison to the other methods tested, and performed similar (in 2D only) to the barycenter method.

We further added a detailed table (**available at Supplementary Information A, pages 8-10 and Fig.S2-S4**) that qualitatively compares the approaches and underlying rationale for the distinct tools that have been benchmarked to provide another level of information on likely advantages and disadvantages of each tool which will further serve the reader.

Table S1 below (Supplementary Information page 7) summarizes the outcome of the benchmarking analyses.

Table S1 Benchmarking analysis. Outcome of accuracy and error percentage based on overlap of polarity values for polar and apolar Cdc42 2D cell layers.

Method	accuracy(%)	
dipole moment method	98.7%	
barycenter method	98.7%	
barycenter method with normalization	98.7%	
intensity profile fit method	94.9%	
ratio method	73.4%	
Fourier method	69.6%	
PCA method	69.6%	
2D neural network	97.4%	

The detailed benchmarking data comparison is shown below (**Supplementary Information A, pages 5 to 6, Fig.S1**)

Fig.S1. Benchmarking analysis. **a)** data set for validation. 2D layers of Cdc42 distribution in individual HSCs, 37 polar and 42 apolar HSCs. **b)-j)** Benchmarking analyses. **b)-h)** Count histograms are shown for the polarity outcome of different methods, with polar Cdc42 2D layer inputs shown in blue and apolar Cdc42 2D layer inputs shown in red. The x-axes differ due to different methods used for analysis of polarity. **b)** Dipole moment method (tool: CellDetail). **c)** Center of mass method/barycenter method. **d)** Center of mass method/barycenter method with normalization. **e)** Intensity profile fit (tool: POME). **b')**

Dipole moment method (tool: CellDetail) with adapted mask to render data compatible for the subsequent benchmarking against the tool QuantifyPolarity. The tool QuantifyPolarity (that can use a Ratio method, a Fourier method or a PCA method) requires a cell mask that excludes the cytoplasm. It was necessary to adapt the input data by inserting a background pixel in the middle of the cell masks used and let it run through CellDetail / dipole moment method again to ensure consistency for comparing results with QuantifyPolarity. **f)** Ratio method (tool: QuantifyPolarity). **g)** Fourier method (tool: QuantifyPolarity). **h)** PCA method (tool: QuantifyPolarity). **i)** Neural network: Inverse cumulative frequency of probabilities over probability outcome for training data plot for determining confidence interval (limits to assignment probabilities). Red markers show misclassified Cdc42 2D cell layers of the training data. Cells falling within the confidence interval (green box) are undefined. **j)** Prediction confusion plot with percentage of true polar, false polar, false apolar and true apolar assigned polar and apolar Cdc42 2D cell layers.

2) While I find the chemistry-inspired dipole moment as a metric of polarity intriguing and novel, I have reservations about its implementation in image analysis. Unlike atoms that have unique charges, fluorescence intensity measurements are sensitive to limits of detection and signal saturation.

- i) I would like a more robust explanation of how normalizing total intensity to 1 and normalization of charge by bit-based and half-cell size could render the dipole moment insensitive to these challenges.
- ii) Overall, the authors should better explain what the nine normalization options in CellDetail are good for (SI lines 49-50) and the kind of artifacts they could generate.

3) In the same vein as above, the authors suggest normalizing the distance between charge centers by maximal diameter to compensate for differences in cell size. However, this normalization would cause the dipole moment to be more sensitive to cell shape. For instance, the maximal diameter of a square is larger than that of a hexagon of the same area, and thus, even if a protein is entirely localized to a single vertex in both cells, the resulting normalized dipole moments will be different. This may not be a problem when analyzing isolated round cells like HSCs, but the application of CellDetail to other cells could be quite limited. In fact, I would like to see a comparison of dipole moment for Cdc42, Tubulin, and Septins versus cell circularity for HSCs to determine if any correlation exists.

Response: These are indeed important questions centering all around the robustness of CellDetail. First, based on the comments of this reviewer, we performed additional analyses to determine how normalizations might influence the robustness of CellDetail. Based on these observations, we decided to set CellDetail to normalize the average intensity to one, which resulted in more robustness of P_n and which puts the value of P_n into the 10^{-6} range (**Fig.1**). We further added succinct explanations within the manuscript with respect to **Fig.1** to explain the benefit of the dipole moment method and the necessity for normalization.

Second, we improved the presentation and explanation of the normalization options and focus on now 7 central and thus critical normalization options (**Supplemental Information E, page 21-28, Table S2 and Fig.S9-S11, also below**). We provide visual guidelines (decisions trees) for the normalization choices and explain them in a more succinct fashion. We further provide a short table summarizing normalization choices with short explanations that directly reflect language of the GUI. The combination of a visual guideline together with a table will serve the reader/user.

Table S2 Normalization options. All implemented normalization options of CellDetail.

Normalization options	
average to 1	'Cell average intensity normalized to 1'
none	'average to 1': unchecked
q_{n1}	'1) Charge normalized dependent on volume of cell (bit)'
q_{n2}	'2) Charge normalized dependent on protein distribution of cell (bit)'
q_{n3}	'3) Cell charge normalized corresponding to maximal value of charge multiplied by number of positive charges'
q_{n4}	'4) Cell charge normalized corresponding to maximal value of charge multiplied by halved number of total charges'
q_{n6}	'6) Own max for each cell (else: dependent on bit)'
R_{n1}	'1) Normalized by maximal diameter'
R_{n2}	'2) Normalized by averaged diameter'

1. Question: Normalization of total intensity per cell

Does the total intensity among cells highly vary?

Fig. S9 Normalization choices. 1. Normalization of total intensity per cell.

2. Question: Normalizations for the type of intensity distribution within a cell

Will you compare polarity between/among one or more biomolecules within a cell, then using different fluorescence channels?

Does a biomolecule of interest vary a lot in its type of spatial distribution within the cells to be analysed?

comparability of biomolecule polarities vs. reward of efficient clustering

3. Question: Normalization of charge

Do images have the same bit depth?

Do single clusters within the same cell have roughly the same intensity values?

same bit depth of images vs not same bit depth of images homogeneous cluster intensities vs. doesn't matter

4. Question: Normalization of distance

How is your biomolecule of interest distributed?

- within protrusions/elongations/uropods? (1)
- more to middle of cell? (2)
- everywhere? (3)

14: R_{n1}

15: R_{n1}

24: R_{n2}

How is the morphology of your cells of interest?

- Anisotropic morphology? (4)
- Isotropic morphology? (5)

25: R_{n2}, R_{n1}

34: R_{n1}

35: R_{n1}, R_{n2}

Fig. S11 Normalization choices.

3. Normalization of charge.

4. Normalization of distance between charge centers.

Third, we performed additional robustness analyses for CellDetail: (i) brightness variation of cells (multiplication till exceeding of maximal value defined by bit depth), (ii) SNR variation (by addition of white Gaussian noise), (iii) oversaturation (by multiplication of cell voxel values and when exceeding the maximal value defined by bit depth, replacing the voxel value with the maximal value defined by bit depth), (iv) shape changes (analytical resolving of diameter at fixed area for different geometric shapes and calculation of corresponding P_n value for one charge sitting at the outmost part of the maximal diameter) (**Supplementary Information B, pages 11-13, Fig.S5**). Overall, CellDetail performs robust with respect to the parameters tested, as none of the cells tested crossed over the trajectory of another cell in any of the tests performed, even upon extreme oversaturation of pixels.

Fig.S5. Analyses of robustness of Cell Detail. **a-d)** on Cdc42 3D IF stacks. **(a)** Image stacks of five HSCs stained for Cdc42. P_n values of these cells cover the range of polarity values obtained for HSCs. Images were used for the analyses in (b-e). **(b)** Brightness variation test outcome for evaluating the influence of intensity variations on the polarity outcome. The polarity parameter P_n is shown against different intensity multiplication factors. **(c)** Signal-to-noise (SNR) ratio test outcome for evaluating the influence of noise on the polarity outcome. P_n is shown against the SNR. **(d)** Oversaturation test outcome. P_n is shown against the percentage of oversaturated pixels. **(e)** Influence of shape test outcome. The percentage difference of P_n for two different shapes of same area size with a charge set at the maximal distance on the outer edge of the maximal diameter are shown against area. The area in pixel count that one layer of hematopoietic stem cells can have is indicated as yellowish box.

4) Unlike other previously published metrics of polarity, the dipole moment is difficult to interpret and unintuitive. That is, the dipole moment values throughout the experiments range from $10e-10$ to $10e-13$, and a higher number does not necessarily mean more polarized localization when comparing two different proteins. For example, it's difficult to judge whether Cdc42 and Tubulin are polarized from the average value of their dipole moments alone. The fact that the average dipole moment for the same protein (and I assume from the same sample) can change by an order of magnitude going from a confocal to a widefield microscope is unsettling. Moreover, Cdc42 and Tubulin are unarguably polarized in young HSCs and not in old HSCs based on the images, but the statistical difference between their dipole moments is rather modest. Here, visual inspection of images seems more informative than the quantifications in Figure 2.a. I would like to see the calculated dipole moments for the images in Fig.2a. printed above each image, as I wonder how these values compare with the averages shown on the box plots.

Response: We agree with the reviewer that it will take time to become more familiar with the metric of the dipole moment. First, based on the comments of this reviewer, we performed additional analyses to determine how normalizations might influence the robustness of CellDetail. Based on these observations, we decided to set CellDetail to normalize the average intensity to one, which resulted in more robustness of P_n and which puts the value of P_n into the 10^{-6} range, for both widefield and confocal as well as for 2D and 3D analyses (**Fig.1**). We further show in **Fig.1e and right** an example on how P_n is actually calculated.

$$\begin{aligned}
 \text{i} \quad q_{n1} &= \frac{\sum q_{pos}}{0.5 \cdot n_{voxel} \cdot ((2^{bit} - 1) - i_{bg}) - i_{av}} = \\
 &= \frac{605.22}{0.5 \cdot 4787 \cdot ((65536 - 1) - 165.87) - 1} \approx 3.87 \cdot 10^{-6} \\
 \text{ii} \quad R_{n1} &= \frac{d_{R+R-}}{\text{maximal diameter}} = \frac{2797.3}{9324.3} \approx 0.3 \\
 \text{i+ii} \quad P_n &= q_{n1} \cdot R_{n1} \approx 3.87 \cdot 10^{-6} \cdot 0.3 \approx \mathbf{1.16 \cdot 10^{-6}}
 \end{aligned}$$

Fig.1e Example calculation of P_n for the protein Cdc42 in HSC.

We further agree with the reviewer that visual images shown with the directly corresponding dipole moment/polarity values will help the reader to get more familiar with the metric dipole moment. As suggested by the reviewer, we now show the according P_n values below the single layer images of the z-stack image in Fig. 2a,b. We now provide in **Fig.2c and below** alignments of P_n values with visual information (aka 3D IF images) along distinct P_n values for both Cdc42 and Tubulin.

In summary, we believe that this novel information will allow the reader to align visual information with the calculated dipole moment and to feel more comfortable with P_n as a novel metric of polarity/distribution.

Fig.2c Example z-layers of individual cells (3D) representing distinct P_n values of Cdc42 and Tubulin.

5) How three-dimensional information of protein localization in the cell can be incorporated into the polarity measurement of CellDetail is unclear.

The authors stated that CellDetail can work with both z-stacked confocal images and widefield images (line 190). However, there are no clear examples of z-stacked image measurement or discussion of how cell orientation and differences in X/Y and Z axial resolutions can affect the dipole moment. In fact, I believe I can see the differences in axial resolution in the supplemental videos, so it would be good to quantify if there is any correlation between the measured dipole moments and the Z axis in these z-stacks (e.g., similar to the analysis presented on Fig.6b).

Response: CellDetail uses 3D information to determine polarity, which sets it also apart from a larger number of tools that can only use 2D information to determine polarity. We show now z-stack images in **Figs.1g and 2c**. How does the 3D information on protein localization result in a polarity measurement? For the analysis, the z-stack tiff files are imported per single cell as a 3D matrix of values. The cell detection and analysis is performed on this 3D matrix. Both the positive and the negative charge-weighted barycenter is calculated in 3D. The distance vector between these two charge centers is multiplied with the absolute value of positive charge, resulting in the dipole moment vector (in 3D). All the data in the manuscript has been acquired with a 500 nm step size of z. Cell orientation and differences in X/Y and Z optical resolutions might indeed affect P_n . In a setting in which the z optical resolution is significantly lower, the definiteness of the z-component of the dipole moment vector is somewhat weaker, and P_n might be a small bit more variable among cells in comparison to images taken at an isotropic resolution. As the lower definiteness does not show directionality of the bias, the effect can result in a slightly elevated spread of the P_n values among cells, while the effect will be small to negligible in cases in which larger numbers of cells are being analyzed like in our analyses. For example, P_n values are very similar for the widefield and the confocal data of Cdc42 and tubulin, while the underlying images that data is obtained from do vary in axial resolution (**Figs.2a,b**)

Minor comments:

6) The aspect ratio of Figure 3 is unsuccessful, and it would be problematic in the PDF version of the manuscript when published. Please consider splitting this figure into two and rebalancing the size of the different panels. For example, the box plots are much smaller than the cartoons of the polarized cells in panels d, e, and i, and it is not easy to see the quantitative differences between dipole moments, which I believe is the key finding of this analysis. Moreover, the line graph in panel c seems unnecessarily big for the information the authors seek to convey with this graph.

Response: We re-organized **Fig.3** to better balance the visual presentation of our data.

7) The methods section should list the filters used to separate the 7-color staining in Figure 5. Also, Figure 5 would be more effective if each channel were shown as a heatmap to reassure readers that the steps taken to prevent bleed-through between channels did not cause the fluorescence in a particular channel to be much lower than the others.

Response: They are now listed in the legend to the previous **Fig.5**, which is now **Supplemental Information C**), but also in Material and Methods: "For the 7 color IF measurements, the filters Semrock FF01-595/31, Semrock FF01-676/29, Zeiss PBP 425/30+514/31+592/25+681/45+785/38, Zeiss QBP 425/30+524/51+634/38+785/38 and Zeiss TBP 467/24+555/25+687/145 were used with the beam splitters TBS 450+538+610, QBS 405+493+611+762, PBS 405+493+575+654+761, BS573, BS652 and with Zeiss Colibri 7 LEDs (385 nm LED, 430 nm LED, 475 nm LED, 555 nm LED, 590 nm LED, 630 nm LED, 735 nm LED)". There are indeed moderate variations in fluorescence intensities among channels, for example due to the limited sensitivity of the emission detector (like for AF790) or indeed with respect to reduce

the risk of spectral overlap (like for example AF405, AF488). Our robustness analysis shown above (**Supplemental information B**) strongly supports that P_n is quite robust with respect to changes in the intensity of the signal.

8) The use of box plots for comparing dipole moments may not be the most effective way to display the data. The difference in the average dipole moments between comparison groups in Fig. 2, Fig. 3, and Fig. 4 is modest. On the other hand, the distribution levels seem to be changing more dramatically between young and old HSCs, and density curves or histograms would make the visual comparisons much easier.

Response: We improved the visualization of the difference of P_n values among young and old and old rejuvenated Cdc42 HSCs by including density curves as well as cumulative curves for P_n (**Fig. 2d,e and below**).

Fig.2. (d) Percentage of cells over their P_n for young, old and old HSCs treated with a pharmacological inhibitor of Cdc42 activity, CASIN (at $5\mu\text{M}$) ($n_{\text{young}} = 92$, $n_{\text{old}} = 98$, $n_{\text{oldCASIN}} = 96$, median $P_{n,\text{young}} = 0.81\text{e-}6$, $P_{n,\text{old}} = 0.64\text{e-}6$, $P_{n,\text{oldCASIN}} = 0.84\text{e-}6$, two-sided Wilcoxon-Ranksum test with $p < 0.05$ for young vs. old and old with CASIN vs. old. $D_{\text{young,old}} = 0.38$, $D_{\text{old,old treated}} = 0.36$) **(e)** Cumulative percentage of HSCs in dependence on P_n value distribution (data from (d)). Setting a previously published level of 62% of young HSC being polar for Cdc42 (by visual determination) as binary separation point for polarity (which also corresponds to a P_n of about $0.6\text{e-}6$ (**Fig.1h**)), 49% of old HSCs are considered polar and 68% of old HSCs treated with CASIN are considered polar.

9) Please ensure that the corresponding p-value for asterisks in all figures is described on the legend for each figure (e.g., the asterisks in Fig. 2.a.i, Fig. 2.a.ii, Fig. 3.b, etc.)

Response: We added all the corresponding p-values to the legends.

Reviewer #2:

I co-reviewed this manuscript with one of the reviewers who provided the listed reports as part of the Nature Communications initiative to facilitate training in peer review and appropriate recognition for co-reviewers.

Reviewer #3:

The ms presents an algorithm for quantification of polarity in the spatial distribution of molecular

concentrations in cell microscopy images. The authors package the algorithm in a self contained computer code with a graphical user interface. The applicability of the algorithm is subsequently demonstrated in the analysis of cytoskeletal protein distributions in hematopoietic stem cells and in fibroblasts.

Over the past decades, classifications of cellular phenotypes have undergone a much-needed shift from the manual qualitative and subjective descriptions to reproducible automated quantifications. As such, the goal of the submitted ms is laudable and useful.

However, in my opinion the tool and the underlying analysis presented in the ms fall short of the requirements for innovation, quality and general-interest that are expected from a Nature Communication publication. Essentially, the tool calculates the first moment of the protein concentration distributions. The authors place considerable emphasis on recasting the concentration values by subtracting the cell-averaged concentration, thereby creating regions of “negative” and “positive” concentrations.

However, I do not see how this provides any extra information, as claimed by the authors. Moreover, multiple numerical quantification schemes for polarity of protein concentrations have been proposed in utilized in the past: Fourier analysis [Merkel et al 2014, Banerjee et al 2017,...]; PCA [Ee Tan et al 2021 Quantify Polarity]; relating concentration polarity to cell edge feature [Farrell et al 2017 SEGGA]; and many others. The authors mention that other schemes suffer from limitations, but since they do not evaluate the benefits of their simplistic approach relative to other procedures, the reader cannot assess this claim.

Response: We performed a large set of benchmarking analyses (**Supplementary Information A, pages 3 to 8, Fig.S1, Table S1**). The benchmarking was performed against the existing methods i) barycenter method with/without normalization, ii) intensity profile fit method, iii) intensity ratio method, iv) Fourier method, v) PCA method, vi) neural network, and used 2D layers of cells visually selected for polarity and apolarity for Cdc42 as the images to be analyzed. As most of the other tools work only on/in 2D, a truly comparative benchmarking analysis could only be performed on 2D images. CellDetail performed much better in separating image layers of HSCs visually scored for being polar or apolar for Cdc42 for all other methods tested, and performed similar (in 2D only) to the barycenter method.

We further added a detailed table (**available at Supplementary Information A, pages 8-10 and Fig.S2-S4**) that qualitatively compares the approaches and underlying rationale for the distinct tools that have been benchmarked to provide another level of information on likely advantages and disadvantages of each tool which will further serve the reader.

Table S1 below (Supplementary Information page 7) summarizes the outcome of the benchmarking analyses.

Table S1 Benchmarking analysis. Outcome of accuracy and error percentage based on overlap of polarity values for polar and apolar Cdc42 2D cell layers.

Method	accuracy(%)
dipole moment method	98.7%
barycenter method	98.7%
barycenter method with normalization	98.7%
intensity profile fit method	94.9%
ratio method	73.4%
Fourier method	69.6%
PCA method	69.6%
2D neural network	97.4%

The detailed benchmarking data comparison is shown below (**Supplementary Information A, pages 5-6, Fig.S1**)

Fig.S1. Benchmarking analysis. **a)** data set for validation. 2D layers of Cdc42 distribution from individual cells, 37 polar and 42 apolar cells. **b)-j)** Benchmarking analyses. **b)-h)** Count histograms are shown for the polarity outcome of different methods, with polar Cdc42 2D layer inputs shown in blue and apolar Cdc42 2D layer inputs shown in red. The x-axes differ due to different methods used for analysis of polarity. **b)** Dipole moment method (tool: CellDetail). **c)** Center of mass/Barycenter method. **d)** Center of mass/Barycenter method with normalization. **e)** Intensity profile fit (tool: POME). **b')** Dipole moment method (tool: CellDetail)

with adapted mask to render data compatible for the subsequent benchmarking against the tool QuantifyPolarity. The tool QuantifyPolarity (that can use a Ratio method, a Fourier method or a PCA method) requires a cell mask that excludes the cytoplasm. It was necessary to adapt the input data by inserting a background pixel in the middle of the cell masks used and let it run through CellDetail / dipole moment method again to ensure consistency for comparing results with QuantifyPolarity. **f)** Ratio method (tool: QuantifyPolarity). **g)** Fourier method (tool: QuantifyPolarity). **h)** PCA method (tool: QuantifyPolarity). **i)** Neural network: Inverse cumulative frequency of probabilities over probability outcome for training data plot for determining confidence interval (limits to assignment probabilities). Red markers show misclassified Cdc42 2D cell layers of the training data. Cells falling within the confidence interval (green box) are undefined. **j)** Prediction confusion plot with percentage of true polar, false polar, false apolar and true apolar assigned polar and apolar Cdc42 2D cell layers.

Finally, I found the paper to be overly verbose and with poor, often unintelligible, use of language.

Response: We improved the wording throughout the manuscript on multiple levels to further improve the experience for the reader.

In conclusion, I do not believe that the presented tool goes further than what is available from multiple available plug-ins for common image processing tools, or the procedures described in the methods sections of numerous publications. The application of the analysis for cytoskeletal protein distributions in HSC and fibroblasts does not provide any significant new results or insights.

Response: We politely disagree with this statement of the reviewer. **i)** The benchmark analysis demonstrates that CellDetail goes farther than current tools, plus, on top, it analyzes polarity and types of distribution in 3D and provides continuous values on distribution, which allows for quantitative correlations that are required to determine changes in a network like the Septins.

ii) The analysis of the Septin network in HSCs and fibroblasts and their change upon aging and senescence does provide significant new insights in the type of change of the Septin cytoskeleton upon aging and senescence. Much like the intermediate filaments, the presence of heteropolymeric networks consisting of multiple Septin isoforms with potential redundancy and the low number of modulators

Fig. 3 **(h)** IF images (examples) of one z-stack plane of old or old HSC treated with CASIN (5μM) (scale bar 5 μm) of Cdc42, Septin6, Septin7 and Septin9. **(i)** P_n of Cdc42, Septin6, Septin7 and Septin9. $n_{\text{old}}=57$, $n_{\text{old,CASIN}}=60$ HSCs from individual mice. Median, two-sided Wilcoxon-Ranksum test, $p < 0.05$ for all proteins analyzed. $D_{\text{Cdc42}} = 0.36$, $D_{\text{Septin6}}=0.34$, $D_{\text{Septin7}}=0.30$, $D_{\text{Septin9}}=0.31$. **(j)** Spearman correlation coefficients R_{Spearman} of P_{ns} of Cdc42 and P_{ns} of Septins of old and old+CASIN HSCs. $n_{\text{old}}=57$, $n_{\text{old,CASIN}}=60$ HSCs from individual mice. $p < 0.05$ for all correlations, except for Septin6 and Septin7 vs. Cdc42 in old HSCs.

have so far slowed down progress of Septin research. CellDetail provides a novel way to analyze heteropolymeric networks, and we provide in the MS a modulator (CASIN) of the network. CellDetail further allows quantification of levels of repolarization of Cdc42 in old HSCs upon CASIN treatment in unprecedented detail (**Fig.2d,e**). We further show novel data that pharmacological attenuation of Cdc42 activity by CASIN, that results in functional rejuvenation of HSCs (Amoah et al., 2022; Florian et al., 2012), affects the polar distribution of Septin6, Septin7 and Septin9 in old HSCs (**Fig.3h-j and above**). The data strongly supports the organization of Septins in HSCs to be a target of elevated levels of Cdc42 activity in aged HSCs and likely to be involved in aging and rejuvenation of HSCs. We also show that the Septin network is altered in senescent human fibroblasts, so disorganization of Septins might be a more general aging-related phenotype. Our data adds to a growing number of recent reports that identify important roles for Septins in stem cell function (Schuster & Geiger, 2021).

The source code is not readily available. The link provided is for the test images used and for stand-alone executable apps for Mac and Windows. I suppose I could have downloaded the executable and tried to extract the code, but this is not "open source", and I do not run precompiled programs that do not provide their source code.

Response: We apologize for the inconvenience that this reviewer was not able to access the source code easily. To guarantee easy access of the source code to almost all systems, we now provide a github repository with installation files, example data, code as txt file, code as .m Matlab file with according images like logos to run the software in owned Matlab software and video tutorials. It is accessible under <https://github.com/xyq91/CellDetail-TS> .

Reviewer #4

General comments: Schuster et al. designed a platform, named CellDetail, to quantify biomolecule distribution in single cells. The algorithm uses the concept of dipole moment, a physical quantity that represents the distribution of positive and negative charges. This paper adapted it to identify regions of high and low protein density by imaging, and regarded them as positive and negative charges respectively. These measurements were then used to quantify the spatial distribution and degree of polarity of protein networks. Authors showed that this algorithm can detect the previously reported polarity change of Cdc42 and Tubulin in HSCs during ageing. They also reported that the disruption of the Septin network, both in aged HSCs and in senescent human fibroblasts which was included as another ageing model to evaluate CellDetail's capability of measuring protein localization dynamics. Overall, CellDetail seems innovative in its application of physical principles to biological image analysis, offering a tool to evaluate biomolecules in single cells.

Response: We thank the reviewer for this overall very positive evaluation of the innovation within CellDetail.

However, my major concern is that the algorithm is specifically designed to measure the spatial distribution of biomolecules, which could limit its application in a broader scenario.

Response: We respectfully disagree with the reviewer on this point. First, as the dipole moment is a general concept for describing spatial distribution, the applicability of the method is not restricted to biomolecules, but rather to any images. Furthermore, other output parameters can be derived,

like similarity or clustering capability or spatial constriction of polarity e.g. to subcellular regions. Second, determination of the distribution of biomolecules in cells is a quite important concept in all life sciences, which qualifies in our view as a broader scenario.

Although experiments were performed to validate the algorithm, they provide limited new biological insights. It would greatly improve the manuscript by providing more evidence that the quantitative value of polarity determined by CellDetail can be translated into biological processes.

Response: The analysis of the Septin network in HSCs and fibroblasts and their change upon aging and senescence does indeed provide significant new insights in the type of change of the Septin cytoskeleton upon aging and senescence. Much like the intermediate filaments, the presence of heteropolymeric networks consisting of multiple Septin isoforms with potential redundancy and the low number of modulators have so far slowed down progress of Septin research. CellDetail provides a novel way to analyze heteropolymeric networks, and we provide in the MS a modulator of the network. (CASIN). We add now additional novel data that show translation into biological processes.

We present novel data that show that pharmacological attenuation of Cdc42 activity by CASIN, that results in functional rejuvenation of HSCs (Amoah et al., 2022; Florian et al., 2012)), affects the polar distribution of Septin6, Septin7 and Septin9 in old HSCs (**Fig.3h-j and right**). The data strongly supports the organization of Septins in HSCs to be a target of elevated levels of Cdc42 activity in aged HSCs and likely to be involved in aging and rejuvenation of HSCs, identifying novel biological processes. We also show that the Septin network is altered in senescent human fibroblast, so disorganization of Septins might be a more general aging-related phenotype. Our data adds to a growing number of very recent reports that identify critical roles of Septins for stem cell function (Schuster & Geiger, 2021).

Specific comments:

1. The rationale of using a quantitative approach, rather than binary measurement, is that it would

Fig. 3 (h) IF images (examples) of one z-stack plane of old or old HSC treated with CASIN (5 μ M) (scale bar 5 μ m) of Cdc42, Septin6, Septin7 and Septin9. **(i)** P_n of Cdc42, Septin6, Septin7 and Septin9. $n_{old}=57$, $n_{old,CASIN}=60$ HSCs from individual mice. Median, two-sided Wilcoxon-Ranksum test, $p<0.05$ for all proteins analyzed. $D_{Cdc42} = 0.36$, $D_{Septin6}=0.34$, $D_{Septin7}=0.30$, $D_{Septin9}=0.31$. **(j)** Spearman correlation coefficients R_{Spearman} of P_n s of Cdc42 and P_n s of Septins of old and old+CASIN HSCs. $n_{old}=57$, $n_{old,CASIN}=60$ HSCs from individual mice. $p<0.05$ for all correlations, except for Septin6 and Septin7 vs. Cdc42 in old HSCs.

be more sensitive in detecting subtle changes of protein distribution. However, the change of P_n between young vs old HSCs seems very minimal (marginal significance and fold change for Cdc42, and not significant for Tubulin with bright field images). This seems inferior to the previously reported several folds of decrease of polarized HSCs during ageing, reported by the same group. Could the authors elaborate on this?

2. It would be helpful to show some examples where CellDetail can detect a subtle change that is not detectable or is challenging by visual assessment. Can authors test and explain in more scenarios and show how does this method outperform binary measurement?

Response: to 1. We improved the visualization of the difference of the P_n values among for example young and old and old rejuvenated Cdc42 HSCs by including density curves as well as cumulative curves for P_n (**Fig. 2d,e and below**). The data show that CASIN treated aged HSCs follow closely the cumulative frequency curve of young HSCs along the range of P_n values, which implies that all types/levels/states of polarity of aged HSCs respond with an increase in polarization upon CASIN treatment. Such information could not have been obtained by a binary analysis.

Fig. 2. (d) Percentage of cells over their P_n for young, old and old HSCs treated with a pharmacological inhibitor of Cdc42 activity, CASIN (at $5\mu\text{M}$) ($n_{\text{young}} = 92$, $n_{\text{old}} = 98$, $n_{\text{oldCASIN}} = 96$, median $P_{n,\text{young}} = 0.81\text{e-}6$, $P_{n,\text{old}} = 0.64\text{e-}6$, $P_{n,\text{oldCASIN}} = 0.84\text{e-}6$, two-sided Wilcoxon-Ranksum test with $p < 0.05$ for young vs. old and old with CASIN vs. old. $D_{\text{young,old}} = 0.38$, $D_{\text{old,old treated}} = 0.36$) **(e)** Cumulative percentage of HSCs in dependence on P_n value distribution (data from (d)). Setting a previously published level of 62% of young HSC being polar for Cdc42 (by visual determination) as binary separation point for polarity (which also corresponds to a P_n of about 0.6 (**Fig. 1h**)), 49% of old HSCs are considered polar and 68% of old HSCs treated with CASIN are considered polar.

To 2. CellDetail is able to detect subtle changes of the relative distribution of components to each other, as we show with the analysis of coordinated changes in the distribution of Septins upon aging/senescence and upon treatment with CASIN. We for example present novel data that show that pharmacological attenuation of Cdc42 activity by CASIN, that results in functional rejuvenation of HSCs (Amoah et al., 2022; Florian et al., 2012)), affects the polar distribution of Septin6, Septin7 and Septin9 in old HSCs (see again **Fig.3h-j and above**). Furthermore, we could show that CASIN treatment increases the correlation of polarity of Cdc42 with the polarity of single Septins. This network behavior is more easily captured via CellDetail than by any binary method as it is based on correlation analyses possible due to the quantitative nature of P_n .

3. Since P_n is a quantitative value, it would be interesting to know if this value has a correlation with other features/functions of HSCs. For example, does a stem cell with high P_n show strong repopulating ability? What about their cycling status, proliferation etc.? Does the P_n value have

correlations with the immunophenotype of HSCs (e.g. expression levels of stem cell markers/genes)? If one can demonstrate (at least to some extent) that the P_n value is predictive of HSC functions/phenotypes, it would make this algorithm a more powerful tool to study HSC biology.

Response: This is a very interesting comment that has been also discussed extensively among the authors. We also agree with the reviewer that such data would strongly add to the field. Unfortunately, we cannot yet sort viable polar/apolar cells, as this remains currently still an intracellular marker that requires for now fixation of cells to stain for. We tried already a variety of other approaches, none of them successful so far, also due to the fact that it remains a continuous parameter in flow.

4. It is not entirely clear how the value of P_n translates to polar vs apolar states. Is it possible to set a threshold based on P_n and annotate cells as polar vs apolar (vs intermediate)?

Response: It is indeed possible to set a threshold, based on visual selection of polar and apolar HSCs. In **Fig.1g,h** and **right**, we show that such a threshold is around a P_n of 0.6×10^{-6} to annotate cells as polar or apolar. We further provide a cumulative frequency graph (**Fig.2e** and **right**, which is a dataset independent from the one in Fig1g,h) which also allowed us to set a threshold of P_n of 0.6×10^{-6} , based on the reported frequency of polar young HSCs, which is highly consistent with data from **Fig.1h**.

Fig.1: (g) Examples of z-stacks (3D) of HSCs visually apolar and polar for the distribution of the protein Cdc42 in HSCs (out of 14 apolar and 23 polar used in (h)). **(h)** Histogram of P_n values for the protein Cdc42 for 14 HSCs visually apolar for Cdc42 and 23 HSCs visually polar for Cdc42. **Fig. 2: (e)** Cumulative percentage of HSCs in dependence on P_n value distribution (data from (d)). Setting a previously published level of 62% of young HSC being polar for Cdc42 (by visual determination) as binary separation point for polarity (which also corresponds to a P_n of about 0.6 (Figure 1h)), 49% of old HSCs are considered polar and 68% of old HSCs treated with CASIN are considered polar.

5. In their previous work, this group demonstrated a decreased polarity of Cdc42 upon aging and showed that restoring Cdc42 distribution by treating aged HSCs with Casin can also revert the differentiation pattern of aged HSCs to more young-like. It would be helpful to show that this rescue of distribution by Casin can be detected using CellDetail as well.

Response: We provide novel data that demonstrates that the repolarization effect of CASIN on old HSCs is detected by CellDetail (**Fig.2d,e** and **above**).

6. The algorithm reported that the Septin network polarity seems to be altered during ageing.

Could this be achieved by visual assessment? A direct comparison of the two methods would be informative and

7. The authors hypothesized that upon ageing, elevated Cdc42 level alters the localization of the Septin network. They determined Septin protein distributions and their relative distances to Tubulin, but not Cdc42. Since the hypothesis is that Cdc42 affects Septin network, it would be more informative to also show the polarity of Cdc42 together with Septins to see if there is a direct correlation between the two.

Response: It is indeed an excellent suggestion to provide additional biological data that provides significant novel insights into a likely role of Septins downstream of Cdc42 with respect to polarity and organization/ compartmentalization of stem cells. Indeed, attenuation of Cdc42 activity in aged HSCs repolarizes the distribution of Cdc42, Septins 6, 7 and 9, and also positively affects their overall co-polarity (**Fig.3h-j and right**).

A higher frequency of aged HSCs with an apolar distribution of Septin7 has been previously reported by Kandi et al. (Kandi et al., 2021). A change in the frequency of cell visually scored for polarity in a binary fashion can be thus indeed achieved. However, CellDetail provides additional inter-related values among all the Septins (like the distance between positive charge centers etc.) which we could not have obtained with a visual assessment alone. A real power of CellDetail is its ability to provide quantitative and continuous values that can be correlated with distribution data of other molecules to identify how for example whole networks (in our case of Septins) change upon aging or any given treatment

8. Imaging proteins in single cells is often a difficult task, especially for cells with irregular morphology or proteins with low expression level. How does CellDetail deal with challenging images, e.g. suboptimal density, low image quality etc.?

Response: We performed a larger set of robustness analyses for CellDetail: (i) brightness variation of cells (multiplication till exceeding of maximal value defined by bit depth), (ii) SNR variation (by addition of white Gaussian noise), (iii) oversaturation (by multiplication of cell voxel

Fig. 3 (h) IF images (examples) of one z-stack plane of old or old HSC treated with CASIN (5 μ M) (scale bar 5 μ m) of Cdc42, Septin6, Septin7 and Septin9. (i) P_n of Cdc42, Septin6, Septin7 and Septin9. $n_{old}=57$, $n_{old,CASIN}=60$ HSCs from individual mice. Median, two-sided Wilcoxon-Ranksum test, $p<0.05$ for all proteins analyzed. $D_{Cdc42} = 0.36$, $D_{Septin6}=0.34$, $D_{Septin7}=0.30$, $D_{Septin9}=0.31$. (j) Spearman correlation coefficients $R_{Spearman}$ of P_n s of Cdc42 and P_n s of Septins of old and old+CASIN HSCs. $n_{old}=57$, $n_{old,CASIN}=60$ HSCs from individual mice. $p<0.05$ for all correlations, except for Septin6 and Septin7 vs. Cdc42 in old HSCs.

values and when exceeding the maximal value defined by bit depth, replacing the voxel value with the maximal value defined by bit depth), (iv) shape changes (analytical resolving of diameter at fixed area for different geometric shapes and calculation of corresponding P_n value for one charge sitting at the outmost part of the maximal diameter) (**Supplementary Information B, pages 11-13, Fig. S5**). In summary, CellDetail is overall robust with respect to the parameters tested, as none of the cells tested crossed over the trajectory of another cell in any of the tests performed, even upon extreme oversaturation of pixels.

Fig.S5. Analyses of robustness of Cell Detail. **a-d)** on Cdc42 3D IF stacks. **(a)** Image stacks of five HSCs stained for Cdc42. P_n values of these cells cover the range of polarity values obtained for HSCs. Images were used for the analyses in (b-e). **(b)** Brightness variation test outcome for evaluating the influence of intensity variations on the polarity outcome. The polarity parameter P_n is shown against different intensity multiplication factors. **(c)** Signal-to-noise (SNR) ratio test outcome for evaluating the influence of noise on the polarity outcome. P_n is shown against the SNR. **(d)** Oversaturation test outcome. P_n is shown against the percentage of oversaturated pixels. **(e)** Influence of shape test outcome. The percentage difference of P_n for two different shapes of same area size with a charge set at the maximal distance on the outer edge of the maximal diameter are shown against area. The area in pixel count that one layer of hematopoietic stem cells can have is indicated as yellowish box.

Reviewer #5:

Changes in the distribution of intracellular biomolecules have been found in aging and diseases. In

this article, the authors developed an open-source image analysis tool, CellDetail, to quantitatively determine the polarity of molecules inside cells in 2D and 3D. CellDetail uses a physics-motivated algorithm to quantify the intracellular polarity of proteins as a real number P_n . This is new and innovative. The authors used CellDetail to analyze 6 individual Septins within a single HSC and discovered the reduction of Septin organization and polarity within aged HSCs, as well as in senescent human fibroblasts. CellDetail provides useful image analysis tools for a quantitative determination of the spatial distribution of protein-protein network in single cells. However, I do have concerns with the effectiveness of the quantification method and interpretation of the analysis of results. So, it may be suitable for publication in Nature Communication after the following concerns are addressed.

Major concerns:

1. Figure 1. How was the center point M calculated? The authors wrote that the program can be downloaded under <https://cloudstore.uniulm.de/s/CnarsbtJ3gYiWww> However, this link does not seem to work.

Response: We apologize for the inconvenience that this reviewer was not able to access the source code easily. To guarantee easy access of the source code to almost all systems, we now provide a github repository with installation files, example data, code as txt file, code as .m Matlab file with according images like logos to run the software in owned Matlab software and video tutorials. It is accessible under <https://github.com/xyq91/CellDetail-TS> .

The center point M was calculated via adding all cell voxel positions, divided by their number

$$M = \frac{\sum_i^n \begin{pmatrix} x_i \\ y_i \\ z_i \end{pmatrix}}{n}$$

(with M being the final center point of the cell and x_i , y_i and z_i being the according cell voxel values x,y,z of cell voxel i . with number n of cell voxels, the sum starts at $i=1$). Later in the calculations, the switch from voxel positions to nm related values takes place (multiplication by given voxel sizes (nm) and repositioning in middle of voxel).

voxel positions to nm related values takes place (multiplication by given voxel sizes (nm) and repositioning in middle of voxel).

2. It would be helpful to provide some intuitive understanding of the physical mean of P_n . Figure 2. What are the P_n values corresponding to the young and old HSCs shown in Figs. 2a(i)-(ii)? Are these cells representative?

Response: We agree with the reviewer that visual images shown with the directly corresponding dipole/polarity values will help the reader to get more familiar with the metric dipole moment. As suggested by the reviewer, we now show the according P_n values below the single layer images of the z-stack images in **Fig.2a,b**. We now also provide in **Fig.2c** and **above** alignments of P_n values with visual information (aka 3D IF images) along distinct P_n values for both Cdc42 and Tubulin.

Fig. 2c Example z-layers of individual cells (3D) representing distinct P_n values of Cdc42 and Tubulin.

3. When the nuclear position and CDC42/tubulin polarity was compared, did the authors use confocal image data or widefield image data?

Response: Widefield image data was used.

4. Figs 3d and 3e, small distance represents both polar with similar R+ and apolar. Is there any way to separate these two representations? Apolar showed random angles, including both large angles and small angles. This means that small angles could represent either polar with similar R+ or apolar. And large angles also could represent either polar with different R+ or apolar.

Response: For separation of these two representations, it is necessary to exclude the apolar distributions. By plotting the distances as well as the angles against the distance between one positive charge center R+ to the center of cell (M), these two representations can be separated by defining a cut-off at which distance of R+ to the center of cell an apolar distribution is present and exclude the according values (**Fig.3d,f and right**).

Fig. 3 (d) Graphical representations for different spatial distribution situations of two components when analyzing distance d . Small distances between both positive charge centers are obtained for a combination polar/polar on the same side and apolar/apolar, while the biggest distances are obtained for opposite situation of polar/polar. (f) Graphical representations for different spatial distribution situations of two biomolecules when analyzing angle α . Small angles between both positive charge centers are obtained for the combination polar/polar on the same side. Large angles between both positive charge centers are obtained for the combination polar/polar on the opposite side. Random angles between both positive charge centers are obtained in case of the combination apolar/apolar.

5. Figure 2. Pearson correlation coefficients for polarity of Cdc42 and nucleus positioning.... Figure 3. The correlation coefficients between P_n and ζ_+ were between 0.2-0.67 considered correlated. What is the physical meaning of Pearson correlation coefficients and is there a uniform measure of what could be interpreted as correlated or not correlated?

Response: The Pearson correlation test identifies a linear relation between two continuous variables (Spearman correlation test would identify monotonous increasing/decreasing relation). As the hypothesis is that the nucleus position has an immediate influence on the polarity, the Pearson correlation coefficient was chosen (linear relationship). The definition of correlation strength goes from none (no significance in p-values) [0] to weak (0 - 0.3], moderate (0.3-0.5] and strong (0.5-0.7] to very strong (0.7-1]. With the Pearson correlation coefficient comes a p-value that below 0.05 is considered as significant (thus correlation) or not significant (no determined correlation).

In the case of the nucleus positioning vs. polarity, we would expect a strong to very strong correlation if only the nucleus positioning would be decisive for polarity. Another possibility for high values is that the protein itself is involved in nuclear processes. High Pearson correlation coefficient values correlate nucleus positioning and polarity, but do not provide causality. However, the lower correlation value for Tubulin is more important: This value shows that there is reduced correlation of nucleus positioning and polarity of Tubulin. Thus, the nucleus positioning is not a decisive factor for the polarity of Tubulin.

6. Figs. 3k and 4d. Pearson correlation coefficient of intensity channels. How were the correlation coefficients and intensity channels computed?

Response: For this calculation the image stacks of two intensity / fluorescence channels together with the cell mask were taken. The voxel values of channel 1 (e.g. red channel) and the voxel values of channel 2 (e.g. green channel), that were considered as cell voxels, were taken. The Pearson correlation coefficient in terms of co-localization analysis is defined by:

$$PCC = \frac{\sum_i (R_i - \langle R \rangle) \cdot (G_i - \langle G \rangle)}{\sqrt{\sum_i (R_i - \langle R \rangle)^2 \cdot \sum_i (G_i - \langle G \rangle)^2}}$$

with R_i and G_i as intensity values of red and green channel of cell pixel i , $\langle R \rangle$ and $\langle G \rangle$ as mean intensities of cell pixels of red and green channels.

7. Figure 3k, the author stated that “For Septin1-Septin2, Septin1-Septin6, Septin1-Septin7, ... and Septin9-Tubulin significantly different medians were found. Nonetheless, the Pearson correlation coefficients among the Septin proteins and also Tubulin remained similar among young and old HSCs”. The statistics show many septin groups are different between young and old HSCs, yet the author concludes that they remained similar. The conclusions are just not convincing. Are the septins significantly different or not?

Response: The Pearson correlation coefficients of Septins are significantly different. For clarification, we rephrased in the MS on page 10 (“Overall, distinct Septin proteins showed significant, but minor changes among young and old HSCs”) and on pages 11 and 12: “The generally high correlation values for both control FF95 and senescent FF95 imply indeed a tight Septin network structure in both types of cells, with some important changes though upon senescence. There is a strong decrease in the correlation of Septin7-Septin9 to Septin7-Tubulin co-localization upon senescence (**Fig.4c**). There is further a decrease in correlations of Septin7-Septin9 to Septin6/Septin9/Septin11-Tubulin, and to a smaller, but still prominent extent also for correlations of Tubulin linked to Septin6-Septin9 and Septin9-Septin11 interactions. A further decrease is visible in correlations of Septin7-Septin9 to Septin6-Septin11 or Septin7-Septin11.”

8. Figs. 4b-4c, did tubulin also show a trend of change? It is not convincing what is the reason that the authors conclude that tubulin did not change upon senescence while some septins showed change. Figs. 4d-4e. There is a strong decrease in the correlation of septin7-septin9 to 7-tub collocation, what is the cause of this? From the results show in Fig. 4d, 7-9 correlation increase upon senescence (although non-significantly). How do we interpret this result?

Response: In contrast to HSCs and to most of the Septins, Tubulin shows in normal and senescent F95 cells a similar 25th and 75th percentile value of P_n . There is in **Fig.4** (former e,d, now **c**) a strong decrease in correlation of Septin7-Septin9 to Septin7-Tubulin co-localization upon senescence, while interestingly, the “simpler” Septin7-Septin9 correlation increases upon senescence. We can only speculate on the underlying mechanisms and consequences. It is likely that this drop in co-localization and/or realignment with Septin 7 is due to changes in spatial localization of Septin9, as the same pattern occurs, though somewhat weaker, in other combinations of Septin9 like with Septins 6 and 11. It is possible that Septin7-Septin9 fulfill additional tasks outside of the “regular” Septin network. As Septin9 is associated with exocytosis, it

might be that this more profound change in the localization of Septin9 in senescent fibroblasts might be linked to the SASP phenotype of senescent fibroblasts. Additional experiments will be required to test this hypothesis.

9. Figure 6a. It seems that p-value needs to be corrected for multiple comparison test.

Response: For Fig.6a (now **Fig.3c**) the distances between positive charge centers of single Septins were obtained for both young and old HSCs. Afterwards, the distances were assessed for significant differences via a two-sided Wilcoxon-Ranksum test, which tests whether it is equal probable that a random value out of population 1 is bigger or smaller than a random value out of population 2. For clarification, we now wrote “distance d between positive charge centers of proteinA-proteinB”.

References

- Amoah, A., Keller, A., Emini, R., Hoenicka, M., Liebold, A., Vollmer, A., . . . Geiger, H. (2022). Aging of human hematopoietic stem cells is linked to changes in Cdc42 activity. *Haematologica*, *107*(2), 393-402. doi:10.3324/haematol.2020.269670
- Florian, M. C., Dorr, K., Niebel, A., Daria, D., Schrezenmeier, H., Rojewski, M., . . . Geiger, H. (2012). Cdc42 activity regulates hematopoietic stem cell aging and rejuvenation. *Cell Stem Cell*, *10*(5), 520-530. doi:10.1016/j.stem.2012.04.007
- Florian, M. C., Klose, M., Sacma, M., Jablanovic, J., Knudson, L., Nattamai, K. J., . . . Geiger, H. (2018). Aging alters the epigenetic asymmetry of HSC division. *PLoS Biol*, *16*(9), e2003389. doi:10.1371/journal.pbio.2003389
- Florian, M. C., Nattamai, K. J., Dorr, K., Marka, G., Uberle, B., Vas, V., . . . Geiger, H. (2013). A canonical to non-canonical Wnt signalling switch in haematopoietic stem-cell ageing. *Nature*, *503*(7476), 392-396. doi:10.1038/nature12631
- Kandi, R., Senger, K., Grigoryan, A., Soller, K., Sakk, V., Schuster, T., . . . Geiger, H. (2021). Cdc42-Borg4-Septin7 axis regulates HSC polarity and function. *EMBO Rep*, e52931. doi:10.15252/embr.202152931
- Schuster, T., & Geiger, H. (2021). Septins in Stem Cells. *Front Cell Dev Biol*, *9*, 801507. doi:10.3389/fcell.2021.801507

NCOMMS-23-35503C Point-to-Point response

Dear Reviewers,

we are honored by the fact that the revised version of our manuscript addressed the comments of the reviewers. The comments of the reviewers allowed us to significantly strengthen the manuscript. There are additional comments from Reviewer 2 and 3. See below our Point-to-Point response to the remaining comments of Reviewer 2 and 3.

Reviewer #2:

I thank the authors for their efforts to address the previous concerns of the reviewers' comments, and I believe the revision has greatly improved the logic and flow of the manuscript. However, I am still not convinced of the sensitivity and reproducibility of CellDetail.

Response: We thank the reviewer for this very positive evaluation of our efforts to further improve our manuscript on CellDetail.

Major comments:

1) While the benchmarking of CellDetail using Cdc42 in comparison with some other published polarity quantification tools has suggested that CellDetail can accurately quantify Cdc42 polarity (about the same accuracy as several published toolkits), there is limited evidence to suggest that CellDetail can be readily applied to other cell polarity quantification. Without other test cases, it is difficult to convince readers that CellDetail can be used in their own polarity systems.

Response: We added novel data to the manuscript as well as to the point-to-point letter here to further demonstrate that CellDetail can be readily applied to other cell polarity quantifications/problems of quantifications of localizations on fluorescent signals in cells.

Novel data in the manuscript:

In a previous study, we investigated the distribution of chromosomes relative to each other within the nucleus upon aging. Quantification of changes in the position of chromosomes and their homologs in the nucleus upon aging remained at this time challenging. At the end, we could only use homolog distance as an indicator for changes in distribution. Re-analysis of the image data with CellDetail now allowed us to quantify changes in the overall position of chromosome 11 in the nucleus upon aging with a polarity distribution approach. The data demonstrates a

Appendix A: Additional application example, Novel Figure S1: a) Published confocal IF example images of Chromosome 11 staining in young and old murine hematopoietic stem cells. b) Published data for percentage of HSCs with homolog proximity of Chromosome 11, published in (1). c) Cumulative percentage of cells over P_n of Chromosome 11 based on the previously published images, determined by CellDetail. $n_{\text{young}} = 39$, $n_{\text{old}} = 37$, $n_{\text{old,CASIN}} = 36$, median: $P_{n,\text{young}} = 0.63e-6$, $P_{n,\text{old}} = 0.40.e-6$, $P_{n,\text{old,CASIN}} = 0.56e-6$, $p_{\text{young vs. old}} = 0.02$; effect size: $f = 0.35$; $p_{\text{young vs. old with CASIN}} = 0.56$; effect size: $f = 0.46$; $p_{\text{old vs. old with CASIN}} = 0.03$; effect size: $f = 0.35$.

The data demonstrates a

very distinct distribution of chromosome 11 in the nucleus of young or aged HSCs, and it confirms that Cdc42 activity inhibition with CASIN restores a youthful distribution of chromosome 11 in the nucleus of murine HSCs. This data serves therefore as an excellent, already visually validated, novel test case for the use of CellDetail for the quantification of distribution of components within cells and will help to convince the reader that CellDetail is a versatile tool that will work in many polarity/distribution systems. We included this data as **Appendix A: Additional application example, Figure S1**.

We also include here in the **point-to-point response additional novel data** on the analysis of the distribution of **histones** in the nucleus of human HSCs. We previously demonstrated that upon aging, certain histone marks are polarly distributed in the nucleus of young and aged murine HSCs. Some of the marks could be visually scored as polar and apolar, while the quantification of changes in the distribution remained difficult. We now use CellDetail to provide a quantitative measure on the distribution of histone marks in the nucleus of young and aged human HSCs. Similar to data in the mouse (1) also in human HSCs, Histone 4 acetylated on lysine 16 (H4K16ac) shows a more polar distribution in young compared to aged human HSCs, which though is not the case for H4K5ac (not polar (most of the cells show a P_n value of smaller than 0.5, and no difference between young and aged human HSCs). This data is work in progress and will form the basis for a novel manuscript on changes of histone marks in aged human HSCs (**Fig.1 point-to-point**, below).

Fig.1 point-to-point. a) Widefield IF example images of H4K16ac staining in young and old CD34+ human hematopoietic stem cells. b) Cumulative percentage of cells over polarity value P_n . The green area indicates a reference window of P_n values at which cells become polar (see Figure 2e, Manuscript). The green lines indicate the P_n value with the largest and the smallest percentage of difference between young and old cells within the green area. $n_{\text{young}} = 167$, $n_{\text{old}} = 71$, median: $P_{n,\text{young}}=0.7\text{e-}6$, $P_{n,\text{old}}=0.6\text{e-}6$ (median) $p = 0.01$; effect size: $D=0.40$ c) Widefield IF example images of H4K5ac staining in young and old CD34+ human hematopoietic stem cells. d) Cumulative percentage of cells over P_n of H4K5ac, $n_{\text{young}} = 155$, $n_{\text{old}} = 93$, median: $P_{n,\text{young}} = 0.33\text{e-}6$, $P_{n,\text{old}} = 0.38\text{e-}6$, $p=0.33$.

2) It is concerning that CellDetail failed to detect the difference in tubulin polarity in young and old HSCs from confocal images, even with the sample size of more than 500 cells and being one of the positive controls that the authors selected (Figure 2a). Can such a difference be picked up by other existing polarity quantification tools? Is that a problem of the tool or the hypothesis? The same goes for Septin1 measurements in young and old HSCs.

In the first submitted version of the manuscript (Figure 3b), there was a significant difference in the polarity quantification output by CellDetail. In the current revision, such difference however is not present anymore (line 281). As a matter of fact, no quantification of Septin1 polarity is included in the main manuscript or supplementary data anymore, prompting me to doubt the authors' confidence in CellDetail themselves.

Response: CellDetail has now shown to be indeed a very robust analysis tool for the determination of differences in polarity/distribution of proteins and other components in cells. While the sample size for the analysis of distribution of tubulin by confocal analysis was only 217 for young and 372 for aged HSCs, it is smaller as stated by the reviewer but still such a large sample size has been so far sufficient to reveal differences (see for example distribution of Septins in fibroblasts or chromosome 11). The transfer of the normalization from individual cell to 1 to average to 1 actually resulted in a more robust analysis as within the first version of the manuscript, as also demonstrated in the robustness tests (**Appendix B benchmarking**, and **Figure 4 point-to-point**, below).

In this light, data on the difference in polarity of Septin 1 is not included in the revised version, as, based on the average to 1 normalization for the data, the difference in distribution of Septin 1 was not significant and we decide to show only significant differences. The most likely explanation of this finding is that there are old cells with a similar level of polarity for Septin 1 as in young cells, which are though at the same time bigger, which likely resulted in a slightly lower frequency of polarity with the previous, volume-dependent normalization method (total to 1). The "new" normalization method (average to 1) is independent of volume and thus presents with more robustness. This analysis also revealed an initial limitation of CellDetail, which we could eliminate in the revised version by a different approach towards normalization. We are thus very confident, also based on the new data added to this point-to-point response, in the ability of CellDetail to quantify polarity and to therefore allow for the determination of differences in polarity among groups of cells.

What might then be an explanation for our seemingly contradictory results (significant difference in widefield and by eye, but not upon confocal imaging)? Widefield microscopy provides images with a lower overall resolution in z-direction as the distinct z-stacks are not acquired with a detection pinhole eliminating light outside of the focus plane so in contrast to confocal imaging, the middle part of the cell with a higher level of overall fluorescence contributes

Figure 2 point-to-point: Analysis of the polarity of tubulin in the 2-3 middle z-stacks per cell.

Confocal imaging:

Analysis of only the 2-3 middle z-stacks of a cell: Median Tubulin: $P_{n,y} = 1.3e-6$, $P_{n,o} = 1.1e-6$, $p < 0.05$, with common language effect size $D_{Tubulin} = 0.43$, $n_y = 217$, $n_o = 372$, two-sided Wilcoxon- Ranksum test.

Analysis of whole cell (data in Figure 3a): Median Tubulin: $P_{n,y} = 1.1e-6$, $P_{n,o} = 1.0e-6$, $p = 0.8$ with common language effect size $D_{Tubulin} = 0.49$

Widefield imaging:

Analysis of only the 2-3 middle z-stacks of a cell: Median Tubulin: $P_{n,y} = 0.64e-6$, $P_{n,o} = 0.59e-6$, $p < 0.05$, with common language effect size $D_{Tubulin} = 0.47$, $n_y = 630$, $n_o = 1370$, two-sided Wilcoxon- Ranksum test.

Analysis of whole cell (data in Figure 3b): Median Tubulin: $P_{n,y} = 0.60e-6$, $P_{n,o} = 0.55e-6$, $p = 0.009$ with common language effect size $D_{Tubulin} = 0.46$.

more to the overall image than the outer range of the cell.

When we focus only on the **2-3 middle layers** of the Tubulin image data from individual cells from the confocal data set as well as the widefield imaging dataset, there is indeed a significant difference in the polar distribution of tubulin between young and aged HSCs (**Figure 2 point-to-point**, right). Adding the upper and lower (relative to the middle) Tubulin filament structures/z-stacks reduces the polarity value in confocal imaging. Evaluation of the middle 2-3 slices of Tubulin image data of the widefield data set shows therefore an even more apolar Tubulin distribution in the old condition as for when evaluating all slices. The difference thus stems from image acquisition technology and the type of distribution of tubulin within cells, which is unique for all the proteins analyzed in the manuscript, as tubulin forms highly filamentous protein structures spanning the whole cytoplasm, like a very complex climbing frame structure on a playground. Thus, **especially for highly filamentous protein components spanning the whole volume of the cell like for example tubulin**, there might be regions of polarity and thus also regions without polarity in individual parts of the cell, on which we tend to then focus on for visual analyses of polarity. We list that as a limitation/point to consider in general for polarity analysis at the end of the manuscript. This data and its interpretation raise of course a couple of additional thoughts and questions. We feel that these are more basic questions on Tubulin biology and thus beyond the current scope of the manuscript, while these analyses demonstrate that CellDetail can contribute to identify and to likely clarify such types of seemingly contradictory results.

3) In the same vein as above, I am less impressed by the application of CellDetail in fibroblast polarity quantification. The two Septins (6 and 11) that exhibit different dipole moment quantifications (Figure 4) are clearly expressed at drastically different levels. Even with all the normalization that the authors included in the CellDetail pipeline, it is still a fair question whether the different readout is because of the low signal in one or the actual polarity difference. I think what is missing here are a positive control to show the sensitivity of CellDetail in less isotropic cells and a direct assessment of how expression level, staining intensity, or signal-to-noise ratio is affecting CellDetail quantification.

Figure 3 point-to-point. Box plots of total intensities aka protein amount of Septins 6,7,9 and 11 in young (CPD 4.3) and aged (CPD 59.0) FF95 fibroblasts. Common language effect sizes: $D_{Sept7} = 0.13$, $D_{Sept9} = 0.24$, $D_{Sept11} = 0.33$. Analyzed were in total 50 young and 68 old FF95s. $p < 0.05$ for all shown Septins except Sept6, two-sided Wilcoxon-Ranksum test.

Response: We determined the overall level of expression (overall pixel intensity) of Septins in young or old FF95 fibroblasts. (**Fig.3 point-to-point**, above). We apologize if the individual cells shown in **Fig.4** implied a strong difference in the level of Septin 6 between young and aged fibroblasts. The level of expression of Septin 6 was similar among young and old fibroblasts, while there were indeed minor differences in overall levels of Septin 7, Septin 9 and Septin 11. The data implies that there is no simple correlation between level of signal/protein and changes in polarity between young and aged FF95 cells, as not only Septin11 shows difference in protein amount, but as well Septin 7 and Septin 9.

To further address the general underlying concerns on robustness of CellDetail with respect to analyzing polarity of Septins in fibroblast, we performed an additional robustness analysis (**Fig.4 point-to-point, side**) to test whether changes in the signal intensity (as implied by the reviewer) might influence polarity quantification, now in fibroblasts. The analysis is similar to what had been already done for HSCs, see **Appendix B benchmarking, Fig. S6**). To this end, brightness for, as an example, Septin 11 was amplified in 4 FF95 cells with distinct levels of P_n without reaching oversaturation of the signal. The additional benchmarking analysis demonstrates that P_n values remained very robust along intensity multiplication factors (aka along changes in signal intensities).

Figure 4 point-to-point. Additional analyses of robustness of CellDetail. Brightness variation test outcome for evaluating the influence of intensity variations on the polarity outcome of Septin 11 in 4 FF95 fibroblasts. The polarity parameter P_n is shown against different intensity multiplication factors.

4) I greatly appreciate the improvement that the authors made to Figure 1 to make the measurement of dipole moments from images much easier to grasp. However, how positive or negative charges (and the subsequent determination of charge centers) are determined is still not clear. I would suggest the authors add such info to Figure 1e, maybe with masks of positive and negative of charges over the fluorescent image on the side and the pinpoint of the charge centers.

$$\begin{aligned}
 \text{i} \quad q_{n1} &= \frac{\sum q_{pos}}{0.5 \cdot n_{vozel} \cdot ((2^{bit} - 1 - i_{bg}) - i_{av})} = \\
 &= \frac{605.22}{0.5 \cdot 4787 \cdot ((65536 - 1 - 165.87) - 1)} \approx 3.87 \cdot 10^{-6} \\
 \text{ii} \quad R_{n1} &= \frac{d_{n-n-}}{\text{maximal diameter}} = \frac{2797.3}{9324.3} \approx 0.3 \\
 \text{i+ii} \quad P_n &= q_{n1} \cdot R_{n1} \approx 3.87 \cdot 10^{-6} \cdot 0.3 \approx 1.16 \cdot 10^{-6}
 \end{aligned}$$

Novel Figure 1e) Example for the calculation of P_n for the protein Cdc42 in HSC (here single 2D layer). Step1 results in the conversion of intensity values into charge values by subtracting the average cell intensity, step2 delivers charge-weighted barycenters based on the equations provided in the Supplementary F: Algorithm. ii) P_n is calculated based on the (i) normalized charge (q_{n1}) and the distance (R_{n1}) between the charge centers.

5) Boxplots in all the polarity measurements are insufficient to represent the distribution of the individual measurement. I would suggest showing all the raw measurements or including the distributions of all the data points in all the quantification output plots.

We agree with the reviewer that boxplots already provide an “aggregated” version of the data. We deliberately decided on using boxplots to present the data, as this serves, in our opinion, the reader well. The information provided by single raw datapoints as shown in the example above does in our opinion not provide additional information compared to the box-plots, and the statistical analysis is not affected by the way the data is presented.

6) No limitation of CellDetail is included in the manuscript. The authors should include a fair assessment of the limitations of the tool in the discussion. Also, there should be a section discussing how to interpret the direct numeric outputs by CellDetail to assess the polarity level.

Response: This is indeed an excellent suggestion. We added, at the end, an assesement on limitations of the tool. Secondly, we also added a section on how to interpret the direct numeric outputs of CellDetail for assessing the polarity level already the first time we introduce P_n .

“It might be best to initially select just a few polar and apolar cells to determine the type and range of the polarity and how well CellDetail separates them. This might help to inform on which normalization optimization to choose if the standard normalization settings do not apply. Depending on the variance of polarity among cells and difference between populations to compare, CellDetail can work already with as few as 15-30 cells to still provide reliable results, but of course, more individual cells will strengthen the analysis. The numeric output value of CellDetail, the normalized dipole moment P_n , remains in itself first an arbitrary order of magnitude. This order of magnitude can also shift slightly, depending on which normalization method was chosen. On the other hand, for the normalization average to 1, as chosen for the data analysis in this manuscript, the numeric data output was quite robust among distinct types of analyses. While there will remain a transition zone for P_n values in which we would list cells positive or negative for being polar based on visual examination, a transition as identified in Figure 2e at around a P_n of 0.6×10^{-6} seems to be a good initial benchmark for calling the distribution polar or apolar, which could even be applied to the distribution of chromosome in the nucleus (**Appendix A**).”

“CellDetail is limited to the analysis of the distribution of components in single cells. As for all IF experiments, the same staining protocols and measurement settings are recommended to allow for a better direct comparison of data and conditions. Low contrast images might lead to underestimation of polarity, which might be adjusted for by choosing the normalized distance (the parameter is called NormalizedDistanceMaxDiameter) instead of the normalized dipole moment for polarity comparison. For the distribution of filamentous proteins like tubulin that form climbing-structure on playgrounds like structure within the cell, distinct levels within the cell might show distinct levels of polarity, which in total might though not present as a polar distribution. In such cases, it might be necessary to carefully consider which level of the cell depicts best the biological function of the network in a given setting. The z-resolution can affect the variability of P_n . In a setting with a low z-resolution, the z-component of the dipole moment vector is somewhat weaker defined, and P_n might be slightly more variable among cells in comparison to images taken at an isotropic resolution.

Figure 5 and data presented in Figure 2b; CellDetail values for the distribution of Cdc42 in young and aged murine HSCs. **Left:** Raw data points, **right:** boxplots as currently within the manuscript.

This might result in a slightly elevated spread of P_n among cells. This effect will be small to negligible in cases in which larger numbers of cells are being analyzed like in our analyses. For example, the spread of P_n values is very similar for the widefield and the confocal data of Cdc42 and tubulin (**Fig.2a,b**), while the underlying images that data is obtained from do indeed vary in axial resolution.”

Reviewer #3:

In the revised version of the ms the authors have gone a long way to address the concerns that I and other reviewers had raised.

Most importantly, the revised ms now thoroughly compares the suggested approach (first moment of the distribution) to other standard polarity quantification schemes in terms of accuracy and robustness. In this regard, the quality of the work is increased significantly.

Response: We thank the reviewer for this very positive evaluation of our efforts to further improve our manuscript.

There is still a lack of a theoretical analysis of this approach, and I maintain that the "dipole moment" terminology is misleading, as there is only one "charge" in the system - but it may be of practical use the the community. My earlier comments regarding the disjointed nature of the septin part stand, however.

Response: We agree with the reviewer that there is a transition within the thought/theoretical process of the underlying methodology in CellDetail. We use the concept of positive and negative charges (dipole moment) for the mathematical equations. The dipole moment is therefore critical to all of our calculations, and by this means integral part of CellDetail, while indeed there is a positive charge (the fluorescent signal) but only a pseudo-negative charge (aka no charge, the absence of the signal). For the underlying biology, treating the distribution of for example a protein like a dipole moment allows us to calculate and quantify distributions of the fluorescent signal relative, using dipole mathematics, to its absence. By this means, we generate a no signal/negative charge reference point within the cell. This duality of the concept (dipole/one charge) is in our opinion actually a/the real strength of CellDetail. This duality allows us to determine distributions and polarity of signals in almost all types of cells and situations, like our new example of the quantification of changes in the distribution of chromosome 11 in the nucleus of aged murine HSCs.

We developed CellDetail to be able to analyze heteropolymeric protein networks, like Septins, in stem and other types of cells, and to investigate in as much aging changes interactions in the network. Our novel data reveal that changes in the Septin network are likely a more general feature of aging cells. Multiple publications support that Septins play a major role in the regulation of cell compartmentalization, and are thus also involved in the regulation of the polar distribution of cell components per se. We regard the analysis of the Septin network in a manuscript focusing on polarity and quantification of polarity as coherent.

Reviewer #3 (Remarks on code availability):

the code is readable and logically organized. Hosting on GitHub was very helpful.

Response: We thank the reviewer for this positive evaluation of our efforts to further improve the accessibility of code.

Reviewer #4 (Remarks to the Author):

In the revised manuscript, the authors have added benchmarking analysis of CellDetail to 2e) their tool's robustness. They also provided more details regarding the methodology behind CellDetail. Using CASIN treatment as a mean to test this tool's ability to detect protein network changes during the rejuvenation of old HSCs, they showed that CellDetail can be used to monitor subtle changes of biomolecule distributions under certain biological process, such as ageing. While it remains unclear if the polarization value (P_n) calculated by CellDetail is linked to specific cell functions due to technical challenges, this revised manuscript has adequately addressed my concerns.

Response: We thank the reviewer for this very positive evaluation of our efforts to further improve our manuscript. We provided in the past indirect evidence that polarity (visually scored) in HSCs is tightly linked to their mode of division. Young, mostly polar HSCs (polar for Cdc42) divide asymmetrically, while aged (mostly apolar HSCs) divided mostly symmetrically (2). P_n values do align with visual scoring of Cdc42 polarity (Figure 2e). P_n values for Cdc42 in HSCs are therefore also tightly linked to the mode of division. Unfortunately, we cannot yet sort viable polar/apolar cells, as this remains currently still an intracellular marker that requires for now fixation of cells to stain for and to be then analyzed by Cell Detail to finally establish a direct correlation of individual values of P_n and the mode of the division of HSCs.

Reviewer #5 (Remarks to the Author):

Thank you for the response. I suggest to accept this paper for publication.

Response: We thank the reviewer for this very positive evaluation of our efforts to further improve our manuscript.

References

1. Grigoryan A, Guidi N, Senger K, Liehr T, Soller K, Marka G, Vollmer A, Markaki Y, Leonhardt H, Buske C, Lipka DB, Plass C, Zheng Y, Mulaw MA, Geiger H, Florian MC. LaminA/C regulates epigenetic and chromatin architecture changes upon aging of hematopoietic stem cells. *Genome Biol.* 2018 Nov 7;19(1):189. Epub 2018/11/09. doi:10.1186/s13059-018-1557-3. Cited in: Pubmed; PMID 30404662.
2. Florian MC, Klose M, Sacma M, Jablanovic J, Knudson L, Nattamai KJ, Marka G, Vollmer A, Soller K, Sakk V, Cabezas-Wallscheid N, Zheng Y, Mulaw MA, Glauche I, Geiger H. Aging alters the epigenetic asymmetry of HSC division. *PLoS Biol.* 2018 Sep;16(9):e2003389. Epub 2018/09/21. doi:10.1371/journal.pbio.2003389. Cited in: Pubmed; PMID 30235201.

NCOMMS-23-35503D Point-to-Point response

There is one comment remaining from reviewer #2.

Reviewer #2 (Remarks to the Author):

Thank you for the response. The additional quantification and explanation is helpful for the interpretation of P_n . I suggest to accept the manuscript for publication if the editors feel adequate.

Response: We thank the reviewer for this positive evaluation of our efforts to further strengthen our manuscript. The additional quantifications and explanations provided will indeed help the reader for understanding the meaning of P_n as a novel parameter to determine distribution of components in cells.

However, this reviewer is still skeptical of the sensitivity and reliability of CellDetail and the potential adaption of this tool by the larger cell polarity community.

Response: We provided along the line of revisions of the manuscript a large number of novel data to demonstrate the robustness (aka reliability) and the sensitivity of analyses performed with CellDetail. This data is primarily provided within the Supplementary Information C and D. We believe that these data demonstrate that CellDetail is indeed a very robust and sensitive analysis tool for the determination of differences in polarity/distribution of proteins and other components in cells.

We believe that CellDetail will be adopted by the scientific community. Treating the distribution of protein like a dipole moment allows us to calculate and quantify distributions of the fluorescent signal relative to its absence. By this means, we generate a no signal/negative charge reference point within the cell. To the best of our knowledge, the difficulties in generating valid reference points for the determination of distributions has been in general hampering approaches to quantify distributions of proteins within cells.

This reference point is in our opinion therefore a/the real strength of CellDetail. It allows us to determine distributions and polarity of signals in almost all types of cells and situations, like our new example of the quantification of changes in the distribution of chromosome 11 in the nucleus of aged murine HSCs. We therefore believe CellDetail to be a more universal tool for the determination of distributions, and that it will for this reason be adopted by the scientific community.